# DEBATE2CREATE: Robot Co-design via Multi-Agent LLM Debate

**Kevin Qiu** [1 2]  **Marek Cygan** [1 3]

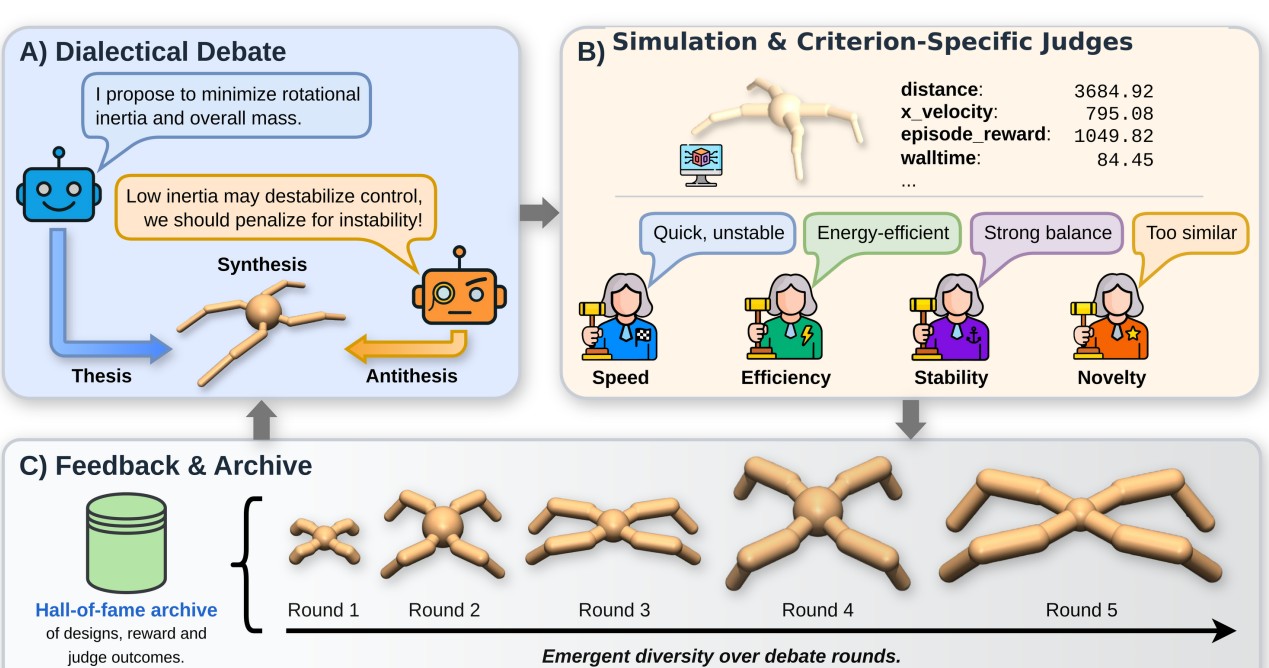

*Figure 1.* Overview of the DEBATE2CREATE framework. (A) A dialectical debate between the *design agent* (🤖) and *control agent* (🤖) to propose and critique design–reward hypotheses. (B) A physics simulator evaluates each proposed design–reward pair, and a panel of criterion-specific judges (👨‍⚖️) reasons over the resulting performance metrics to provide feedback. (C) A hall-of-fame archive stores the best design–reward pairs from each round to inform subsequent debates. **Takeaway:** Structured debate proposes design–reward hypotheses that are selected by simulator score, not LLM preference.

## Abstract

We introduce DEBATE2CREATE (D2C), a multi-agent LLM framework that formulates robot co-design as structured, iterative debate grounded in physics-based evaluation. A *design agent* and *control agent* engage in a thesis–antithesis–synthesis loop, while criterion-specific LLM judges provide multi-objective feedback to steer exploration. Across five MuJoCo locomotion benchmarks, D2C achieves the highest default-normalized score among the evaluated LLM-based and black-box baselines, with gains up to $3.2\times$ on Ant and

nearly $9\times$ on Swimmer. Iterative debate yields 18–35% gains over compute-matched zero-shot generation, and D2C-generated rewards transfer to default morphologies in 4/5 tasks. These results suggest that structured, simulator-grounded multi-agent interaction is a useful mechanism for joint morphology–reward optimization under a fixed-topology, per-candidate-RL protocol. Project page: debate2create.github.io.

## 1. Introduction

A robot's body and its control objective are fundamentally coupled. Lengthen a leg and the optimal gait changes. Tweak the reward to penalize energy and an otherwise agile body crawls. This coupling implies that optimizing morphology under a fixed reward can yield bodies that become fragile when the reward changes, while engineering

[1]University of Warsaw [2]IDEAS NCBR [3]Nomagic. Correspondence to: Kevin Qiu <kevinxqiu@gmail.com>.

*Proceedings of the 43rd International Conference on Machine Learning*, Seoul, South Korea. PMLR 306, 2026. Copyright 2026 by the author(s).

rewards for a fixed body limits achievable performance. The joint search space is high-dimensional and nonlinear, making exhaustive exploration impractical. Yet addressing this challenge is critical as co-designed robots can outperform their fixed-body or fixed-reward counterparts by exploiting body–controller synergies that neither optimization can discover (Sims, 1994).

Recent work has shown that large language models (LLMs) can generate reward code (Ma et al., 2024a;b) or propose morphology edits (Qiu et al., 2026; Ringel et al., 2025), but most existing pipelines treat body and reward as separate problems. This misses opportunities for co-adaptation, where a reward tuned for one morphology may be suboptimal for another. Worse, single-agent generation tends to reuse familiar patterns, narrowing exploration precisely where co-design demands diversity.

We introduce DEBATE2CREATE (D2C), a multi-agent LLM framework that addresses these limitations through structured debate (Figure 1). A *design agent* proposes morphology edits while a *control agent* critiques designs and writes reward code, engaging in a thesis–antithesis–synthesis loop where proposals are refined before evaluation. A panel of criterion-specific LLM judges, each focused on a different objective, provides feedback that steers exploration away from narrow optima. All selection decisions are grounded in physics simulation, where candidates are trained in Brax (Freeman et al., 2021) and ranked by a standardized task score, ensuring that debate rhetoric is governed by empirical performance.

On five MuJoCo locomotion tasks, D2C achieves $3.2\times$ the score of the unmodified Ant morphology and the highest default-normalized score among the evaluated LLM-based and black-box baselines under the same evaluation metric. Cross-over experiments reveal that D2C-generated rewards transfer to default morphologies, improving 4 of 5 tasks without design changes. This suggests the learned shaping captures transferable locomotion principles rather than morphology-specific hacks (Ng et al., 1999).

Our contributions are as follows:

1. **A multi-agent debate framework for co-design.** We introduce D2C, which, to our knowledge, is the first framework that jointly optimizes morphology and reward through a role-separated thesis–antithesis–synthesis debate loop with cross-agent critique.
2. **Criterion-specific LLM judges.** Role-specialized judges provide multi-objective critiques that improve metric coverage and steer exploration toward diverse solutions.
3. **Ablation and transfer experiments.** We evaluate D2C against LLM-based and black-box baselines on MuJoCo locomotion tasks, with cross-over experiments showing

reward transferability to unmodified morphologies and ablations quantifying the 18–35% gains from iterative debate over single-pass generation.

## 2. DEBATE2CREATE

### 2.1. Problem Formulation

We formalize robot co-design as jointly selecting a morphology $m$ (design parameters) and a reward function $r$ that induces a control policy via reinforcement learning (RL). Let $\mathcal{M}$ be the space of feasible morphologies and $\mathcal{R}$ a family of reward functions. For a candidate pair $(m, r) \in \mathcal{M} \times \mathcal{R}$, the optimal policy is

$$\pi^*(m, r) = \arg\max_{\pi} \mathbb{E}_{\pi}\left[\sum_{t=0}^{T} r(s_t, a_t)\right], \qquad (1)$$

where $s_t$ and $a_t$ denote the state and action at time $t$ under policy $\pi$ on morphology $m$, and $T$ is the episode horizon. We denote by $S(m, \pi)$ the task-specific performance score achieved by policy $\pi$ on morphology $m$, measured in simulation. For a candidate $(m, r)$, the score is $S(m, \pi^*(m, r))$. Section 3.1 defines $S$ precisely for our experiments. The goal of co-design is to find:

$$\max_{m \in \mathcal{M}, \, r \in \mathcal{R}} S(m, \pi^*(m, r)). \qquad (2)$$

We approximate this joint optimization by iteratively refining $(m, r)$ through structured debate rounds, where agents propose, critique, and revise candidates before evaluation.

A key consideration is to separate training from evaluation to reduce specification gaming (Amodei et al., 2016; Krakovna et al., 2020). During training of a candidate $(m, r)$, the policy is optimized with reward $r$ to obtain $\pi^*(m, r)$. At evaluation time, all candidates are compared using the same standardized task score $S(\cdot)$, which is interpretable across different designs and controllers.

### 2.2. LLM Agents and Debate Procedure

Our debate procedure follows a thesis–antithesis–synthesis structure: the design agent proposes an initial design (*thesis*), the control agent raises critiques (*antithesis*), and the design agent revises the proposal to address these concerns (*synthesis*). This structure is particularly suited to co-design because morphology and reward interact non-linearly: a body change that improves locomotion under one reward may destabilize the robot under another. By having the control agent critique thesis designs before synthesis, the framework identifies controllability issues early, avoiding expensive simulation of fundamentally unstable designs.

**Design agent ( 🤖 ).** The design agent is prompted as a robot design engineer. At each round, it receives the current

robot design $m$, a description of the task, performance metrics from the most recent simulation (e.g., forward velocity, energy consumption, episode length), and a digest of the hall-of-fame archive containing the top-$k$ design–reward pairs ranked by evaluation score $S$. Based on this context, it proposes a specific edit to $m$ that adjusts exposed design parameters, along with a short rationale explaining why the modification could improve performance.

**Control agent ( ).** The control agent is prompted as a reward function engineer and serves two roles in the debate loop. During the antithesis phase, it critiques thesis designs proposed by the design agent, highlighting potential control challenges or structural weaknesses. During the *reward generation* phase, it receives the revised (synthesis) design $m'$ along with the task description and recent performance metrics, and outputs a reward function $r$ tailored to $m'$ as a code snippet following a predefined template for computing per-timestep rewards. We validate this code via syntax checking and test execution on a single rollout before inserting $r$ into the training loop. Failures trigger self-repair (see below). This approach follows Eureka (Ma et al., 2024a), but the reward is explicitly conditioned on the current morphology.

**Reliability checks.** To ensure robustness to LLM generation errors, we validate every proposed morphology and reward before training. Reward code compiles on the first attempt in 97% of cases, and self-repair (re-prompting with traceback) resolves 99% of failures within two attempts. XML validation rejects <2% of morphology proposals due to constraint violations. Overall, <1% of candidates are discarded. Our self-repair procedure follows prior work on LLM self-debugging (Chen et al., 2023; Shinn et al., 2023; Madaan et al., 2023).

**Criterion-specific judges ( ).** Given a proposed design $m'$ and reward $r$, we instantiate the morphology in simulation and train a policy for a fixed budget of environment interactions. This yields various performance metrics for the design–reward pair, including locomotion progress, stability, and energy efficiency. A panel of four LLM-based judges then analyzes the metrics for the top-scoring candidate in each round. Each judge focuses on a specific criterion (speed, stability, energy efficiency, or novelty) and produces concise textual feedback highlighting strengths and weaknesses of the design under $r$. These critiques are concatenated and provided to both agents in the subsequent round. The goal is to surface multi-criteria trade-offs that help avoid converging to brittle local optima.

**Archive and grounding.** The hall-of-fame archive stores all evaluated design–reward pairs ranked by task score $S$, playing an analogous role to quality-diversity archives (Mouret & Clune, 2015; Pugh et al., 2016). All selection and ranking decisions are based on the simulator task score $S$, while

LLM judges provide feedback only. Unlike typical LLM debate settings where a learned judge arbitrates persuasiveness, here the "judgment" is grounded in physics metrics, providing an objective signal that drives proposals toward validated designs. After completing $K$ debate rounds, we select the highest-scoring design–reward pair from the archive as the final outcome. The output of D2C is an optimized robot morphology (provided as an XML specification) along with the reward function that achieved the highest task score $S$. Algorithm 1 summarizes this procedure.

---

**Algorithm 1** DEBATE2CREATE: Dialectical Co-design Loop.

---

1: **Input:** initial design $m_0$, archive $\mathcal{H} \leftarrow \emptyset$, rounds $K$, samples $N_m$, $N_r$
2: Initialize current best $m^* \leftarrow m_0$
3: **for** $k = 1$ **to** $K$ **do**
4:     **Thesis:** design agent proposes $\mathcal{T}_k = \{m_{k,i}^{(\text{th})}\}_{i=1}^{N_m}$ given $(m^*, \mathcal{H})$
5:     **Antithesis:** control agent critiques each $m \in \mathcal{T}_k$ *(thesis not trained)*
6:     **Synthesis:** design agent revises $\mathcal{T}_k \rightarrow \mathcal{S}_k = \{m_{k,i}^{(\text{syn})}\}_{i=1}^{N_m}$
7:     **for** $i = 1$ **to** $N_m$ **do**
8:         Control agent proposes rewards $\{r_{k,i,j}\}_{j=1}^{N_r}$ for $m_{k,i}^{(\text{syn})}$
9:         **for** $j = 1$ **to** $N_r$ **do**
10:           Train $\pi_{k,i,j}$ on $(m_{k,i}^{(\text{syn})}, r_{k,i,j})$; score $s_{k,i,j} \leftarrow S(m_{k,i}^{(\text{syn})}, \pi_{k,i,j})$
11:         **end for**
12:     **end for**
13:     Update archive: $\mathcal{H} \leftarrow \mathcal{H} \cup \{(m_{k,i}^{(\text{syn})}, r_{k,i,j}, s_{k,i,j})\}_{i,j}$
14:     Round winner: $(m^*, r^*) \leftarrow \arg\max_{(m,r,s) \in \mathcal{H}_k} s$ where $\mathcal{H}_k$ is round-$k$ candidates
15:     **Judge feedback:** criterion-specific judges critique $(m^*, r^*)$
16:     Provide archive digest and critiques to agents for round $k{+}1$
17: **end for**
18: **Output:** $\arg\max_{(m,r,s) \in \mathcal{H}} s$

---

## 3. Experiments

### 3.1. Experimental Setup

**Implementation details.** We use a two-stage evaluation protocol: (1) *search phase*: run D2C for 3 independent seeds per environment to identify the best design–reward pair, (2) *final evaluation*: retrain the best pair 5 times to compute mean $\pm$ std for reporting. Each debate round samples $N_m{=}4$ candidate morphologies and generates $N_r{=}4$ reward variants per morphology, yielding 16 synthesis design–reward candidates per round. Thesis proposals elicit critique, but are not trained. A 5-round run therefore trains $5 \times 16 = 80$ policies per seed and environment. All baselines use identical RL training budgets and evaluation protocols. We instantiate all agents with GPT-5.2 (`gpt-5.2-2025-12-11`). Policy training runs in Brax headlessly on a cluster with 8 NVIDIA Titan V-12GB GPUs,

*Table 1.* Design and reward ablation on five MuJoCo tasks (5 final training seeds). The first two columns indicate the source of morphology and reward ("Default" = original MuJoCo morphology/reward; "D2C" = generated by our method). For each row, the best (morphology, reward) pair is retrained 5 times to compute mean ± std. Entries report task score $S$ (rounded to the nearest integer); best mean per column is in bold. We also report the difference-in-differences interaction term $\Delta_{\text{int}}$ computed from the four mean scores per environment; positive values indicate super-additive co-design synergy beyond morphology-only and reward-only improvements. **Takeaway:** Both morphology and reward contribute to performance; full D2C (bottom row) achieves the best results in 4/5 tasks.

| Morphology | Reward | Ant | HalfCheetah | Hopper | Swimmer | Walker2D |
|---|---|---|---|---|---|---|
| Default | Default | 14886 ($\pm$459) | 21251 ($\pm$1245) | 4183 ($\pm$299) | 926 ($\pm$34) | 9596 ($\pm$429) |
| Default | D2C | 17402 ($\pm$1189) | 22195 ($\pm$222) | 5383 ($\pm$19) | 904 ($\pm$28) | 11027 ($\pm$238) |
| D2C | Default | 42773 ($\pm$2737) | 26266 ($\pm$951) | 3677 ($\pm$274) | **9379** ($\pm$61) | 10552 ($\pm$563) |
| D2C | D2C | **47084** ($\pm$1469) | **28517** ($\pm$1353) | **6257** ($\pm$112) | 8639 ($\pm$63) | **13062** ($\pm$1198) |
| Interaction $\Delta_{\text{int}}$ | | 1795 | 1307 | 1380 | -718 | 1079 |

evaluating 8 candidates in parallel. Appendix B reports RL hyperparameters and Appendix C summarizes the candidate and LLM request budget.

**Normalization.** We normalize each method's score by the Default baseline score from the same seed, reporting mean $\pm$ std of this ratio across seeds (e.g., $3.2\times$ indicates scoring 3.2 times higher than the default).

**Environments.** We evaluate on five locomotion tasks adapted from the MuJoCo benchmark (Todorov et al., 2012): Ant, HalfCheetah, Hopper, Swimmer, and Walker2D. We use the Brax physics simulator (Freeman et al., 2021) with MuJoCo-style XML definitions. Brax provides GPU-accelerated headless simulation that enables parallel policy training, while environment dynamics and default morphologies follow the OpenAI Gym specification (Brockman et al., 2016). For the design agent, we use the official MuJoCo XML models with a subset of parameters exposed for editing (Appendix F lists the editable parameters per environment). For the control agent, the LLM receives the task description and environment code, and reward code is masked out for the agent to propose. We use a single standardized scalar score $S$ for selection across all methods (defined below).

**Baselines.** We compare D2C to four reference methods: **RoboMoRe** (Fang et al., 2025), an LLM-based co-design method using coarse-to-fine refinement (125 candidates); **Eureka** (Ma et al., 2024a), LLM-based reward design on fixed morphology (5 independent search runs); **Bayesian Optimization (BO)** (Snoek et al., 2012; Shahriari et al., 2016), black-box morphology search with fixed reward; and **Default**, the original MuJoCo morphologies with hand-crafted rewards. BO optimizes morphology parameters only. We exclude continuous reward search because reward functions are structured code objects that do not admit a natural continuous parameterization. Eureka uses 5 search seeds compared to D2C's 3; as with the other $n=3$ comparisons, we treat differences as descriptive under the same evaluation metric. Appendix E provides additional baseline details.

**Evaluation.** We train each candidate with a fixed RL budget

using PPO (Schulman et al., 2017) or SAC (Haarnoja et al., 2018) (Appendix B) and score it with $S(m, \pi) = \sum_{t=0}^{T-1} d_t$, where $d_t$ is the 2D distance from the origin at timestep $t$. This metric is *reward-agnostic*: comparing episode returns would conflate task performance with reward engineering choices, so we evaluate all methods on the same external $S$. Appendix H confirms $S$ correlates with standard returns (Spearman $\rho = 0.61$, $p < 0.001$); candidates selected by best $S$ rank in the 96.9th percentile of standard return on average. All methods use identical RL budgets; we report candidate counts explicitly and include a compute-matched zero-shot ablation (Section 3.4).

### 3.2. Main Experiments

**Main claim.** Under our fixed-topology, per-candidate-RL protocol, iterative simulator-grounded debate improves design–reward co-design over single-pass generation on all five tasks and over the evaluated LLM-based and black-box baselines on all five tasks. D2C achieves the highest Default-normalized score on each task under an 80-candidate budget (Figure 2) and outperforms RoboMoRe despite evaluating 36% fewer candidates (80 vs. 125). BO and RoboMoRe produce near-zero scores on Hopper and Walker2D under our protocol; inspection of the resulting XMLs suggests kinematically non-functional morphologies (e.g., BO's best Hopper has a thigh segment of 0.03 units, and the Walker2D leg radii exceed segment lengths). These designs satisfy parameter bounds but often cannot stand, so the controller receives little locomotion signal (Appendix E). D2C's control agent critiques designs before simulation, and synthesis revisions pair morphologies with balance-shaped rewards.

Figure 2 summarizes performance across all five tasks under the same scalar score $S$ for D2C, Eureka, BO, RoboMoRe, and Default. All methods use identical per-candidate RL training budgets and the same evaluation metric; bars report Default-normalized mean $\pm$ std over 3 search seeds. D2C achieves the largest gains on Ant ($3.2\times$) and Swimmer ($\sim 9\times$), while HalfCheetah, Hopper, and Walker2D show smaller but consistent gains ($1.3$–$1.5\times$). Appendix G.1 re-

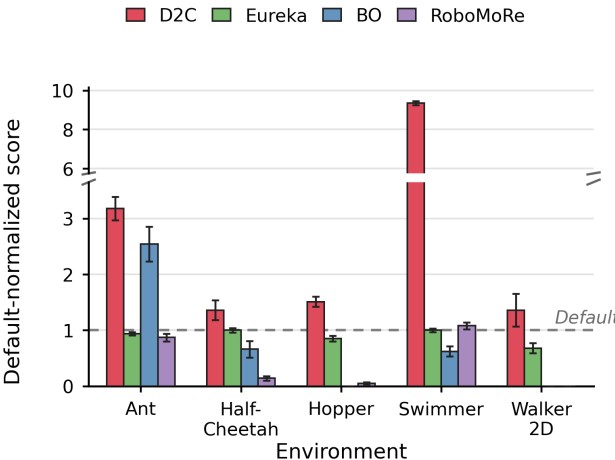

*Figure 2.* Default-normalized performance across environments (mean ± std over 3 search seeds). The y-axis uses a broken scale so the Swimmer gain remains visible without compressing the remaining tasks. **Takeaway:** D2C achieves the highest score across all five evaluated tasks.

ports Welch $t$-tests and Hedges' $g$ effect sizes over the $n=3$ search seeds. Because $n$ is small, we treat these statistics as descriptive summaries and rely primarily on effect sizes and per-seed values rather than formal significance.

**D2C learns transferable reward shaping.** Table 1 isolates morphology and reward contributions. Applying the D2C reward on the default morphology improves performance in 4/5 environments, suggesting the learned reward shaping is not tied to the discovered morphology. This transferability indicates that debate identifies reusable shaping patterns (e.g., stability maintenance, smooth actuation) rather than only morphology-specific rewards. The strongest results arise when both design and reward are produced by D2C, which outperforms the cross-over conditions in all tasks except Swimmer. In Swimmer, morphology changes alone yield ~10× gains, leaving little room for reward shaping due to its low-DOF dynamics.

**Co-design interaction.** Table 1 exhibits non-additive interactions that suggest synergy between morphology and reward. We quantify this via a difference-in-differences term $\Delta_{\text{int}} = S(\text{D2C}, \text{D2C}) - S(\text{D2C}, \text{Def.}) - S(\text{Def.}, \text{D2C}) + S(\text{Def.}, \text{Def.})$. This term is positive in 4/5 tasks (e.g., Hopper: $\Delta_{\text{int}} = 1380$), indicating that D2C's reward shaping is most effective when paired with the morphology it was co-designed for. Swimmer is the exception, consistent with its morphology-dominated gains.

**Per-environment analysis.** Ant shows large gains ($3.2\times$) due to a rich design space where wider stances and longer limbs improve stability at speed. Swimmer achieves the largest gains ($\sim 9\times$) because its low-DOF dynamics make morphology changes disproportionately impactful. HalfCheetah, Hopper, and Walker2D show modest gains

($1.3$–$1.5\times$) as bipedal dynamics impose tighter stability constraints. In these environments, reward shaping contributes a larger fraction of improvement (Table 1).

### 3.3. Debate Dynamics

**D2C improves over debate rounds.** Figure 3 shows best-so-far improvement over rounds across all five environments, along with the Default-normalized baseline (1.0) and a zero-shot reference that generates 80 candidates without iterative feedback. Performance improves over rounds across all tasks, with D2C continuing to improve in later rounds (3–5). Appendix G.3 (Figure 8) illustrates this progression for Ant: the best-performing morphology evolves from a compact design to one with a smaller torso, longer legs, and wider stance. To illustrate the debate dynamics: in Round 0, the stability judge noted "marginal upright stability with increased contact," prompting the design agent to widen the stance ($0.22 \rightarrow 0.30$) and increase torso radius ($0.14 \rightarrow 0.16$), yielding a 33% score improvement in Round 1.

**D2C generates diverse designs.** Figure 4 shows representative final morphologies across tasks. D2C proposes non-trivial edits to limb lengths, torso proportions, and joint placements that reflect task-specific trade-offs (e.g., longer stride vs. stability). These are not simple scaling of the default model. The variety in final designs suggests that debate explores distinct regions of the design space rather than converging to a single canonical shape. In contrast, BO lacks semantic understanding of physical constraints, leading to ~12% invalid morphology proposals compared to <2% for D2C (Appendix E).

**D2C generates structurally distinct reward functions.** Table 1 shows that the learned reward shaping improves performance even on the default morphology, indicating effects beyond morphology changes alone. To characterize structural differences, Figure 5 reports cosine similarity between reward-code feature vectors extracted via AST parsing. We extract term frequency vectors over reward components such as velocity bonuses, stability penalties, and energy costs (see Appendix D for details). Lower similarity for D2C indicates that its rewards use different combinations of shaping motifs than prior methods. Exact D2C reward expressions are provided in Appendix D.

**Common reward shaping patterns.** A post-hoc analysis of reward expressions (Appendix D) reveals recurring motifs. These include bounded nonlinearities to reduce reward hacking, smoothness penalties (4/5 environments) to stabilize novel morphologies, and orientation terms where debate identified balance as critical. Notably, backward-motion penalties emerged after the control agent observed oscillatory behaviors exploiting training reward. These patterns suggest debate can expose failure modes that single-pass generation misses. The consistency of these motifs across

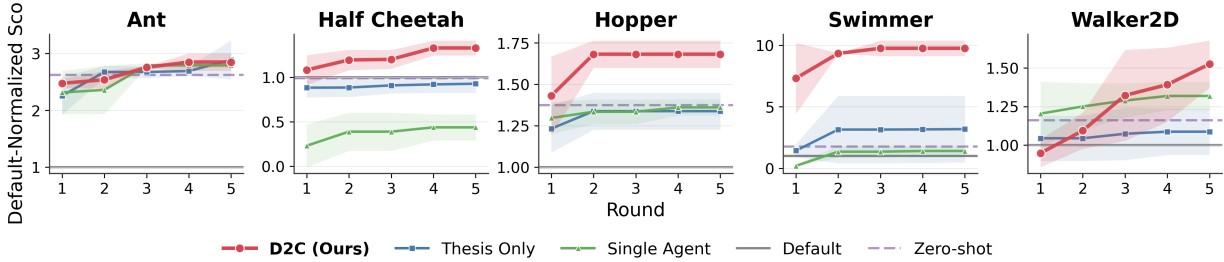

*Figure 3.* Best-so-far Default-normalized scores over debate rounds (mean $\pm$ std over 3 search seeds). Dashed lines: Default baseline (1.0) and zero-shot (80 candidates, no iterative feedback). **Takeaway:** Iterative debate consistently improves over rounds, outperforming zero-shot generation by 18–35%.

environments indicates that debate discovers reusable shaping strategies rather than only environment-specific heuristics.

### 3.4. Ablation Studies

We ablate three key components of D2C: iterative debate (vs. zero-shot), role separation (vs. single-agent), and criterion-specific judges (vs. no judges).

**Iterative debate outperforms zero-shot generation.** As shown in Figure 3, the zero-shot baseline (dashed line) generates 80 candidates without iterative feedback. Despite identical candidate budgets, D2C surpasses the zero-shot optimum in later rounds (3–5), achieving 18–35% higher scores (mean 24%) across environments. This confirms that iterative critique and synthesis guide exploration beyond single-pass generation.

**Role separation and synthesis matter.** We evaluate two additional ablations (Figure 3). In the *single-agent* setting, one LLM proposes both morphology and reward without role specialization or cross-critique, but retains iterative rounds. In the *thesis-only* setting, we disable synthesis and train only initial proposals. Both underperform full D2C: single-agent shows weaker improvement over rounds, while thesis-only plateaus early. These results support the role of specialization and critique-then-revise synthesis in sustaining gains beyond initial proposals.

**Judges influence design decisions.** We analyzed co-occurrence patterns between critique themes and parameter changes (Appendix G.2). Chi-squared tests reveal significant associations ($p < 0.001$): speed critiques predict leg geometry changes, stability critiques predict leg and torso modifications. Runs with criterion-specific judges show smaller parameter change magnitudes than without judges ($0.126 \pm 0.102$ vs. $0.220 \pm 0.134$), suggesting more targeted refinements.

**Criterion-specific judges improve multi-objective coverage.** Figure 6 reports Pareto hypervolume (Zitzler & Thiele, 1998; Zitzler et al., 2003). In environments with multi-

dimensional trade-offs (Ant, Hopper, Walker2D), we compute 3D hypervolume over distance, stability, and efficiency, observing that criterion-specific judges improve coverage in all three. In HalfCheetah and Swimmer, the environments are inherently upright, so stability is less informative and hypervolume reduces to 2D. In these environments, the stability judge still provides feedback that does not influence outcomes, which may dilute useful critiques. A fairer comparison would remove the stability judge entirely for inherently stable tasks.

**LLM backbone sensitivity.** We evaluated three GPT-5 scales (nano, mini, full) on Ant to assess whether D2C's gains depend on LLM capability. All models achieve comparable final archived scores ($3.18$–$3.60\times$ normalized; pairwise $t$-tests $p > 0.19$), suggesting that performance stems from the debate framework rather than raw model scale. However, smaller models produce substantially more degenerate candidates: GPT-5-mini has a $49.2\%$ failure rate (near-zero scores) versus $15.8\%$ for GPT-5.2. The archive mechanism makes D2C robust to individual failures, but smaller models waste computation on unviable designs.

**Open-weight backbones.** Appendix J extends this analysis to two DeepSeek open-weight backbones on Ant and Swimmer under the same 3-seed, 80-candidate protocol. DeepSeek-R1 reaches $2.47\times$ and $6.62\times$ the Default baseline on Ant and Swimmer, while DeepSeek-V3 reaches $1.85\times$ and $3.72\times$, indicating that the debate loop remains useful without the proprietary main backbone, although absolute performance still trails GPT-5.2.

## 4. Discussion

**What the debate structure contributes.** The strongest evidence for D2C is not only that its final scores are higher, but that the improvement pattern matches the intended mechanism. A compute-matched zero-shot baseline evaluates the same 80 candidates without iterative feedback and remains 18–35% below D2C, while single-agent and thesis-only ablations plateau earlier. The cross-over study in Table 1 shows why this matters for co-design: in Hopper, the D2C

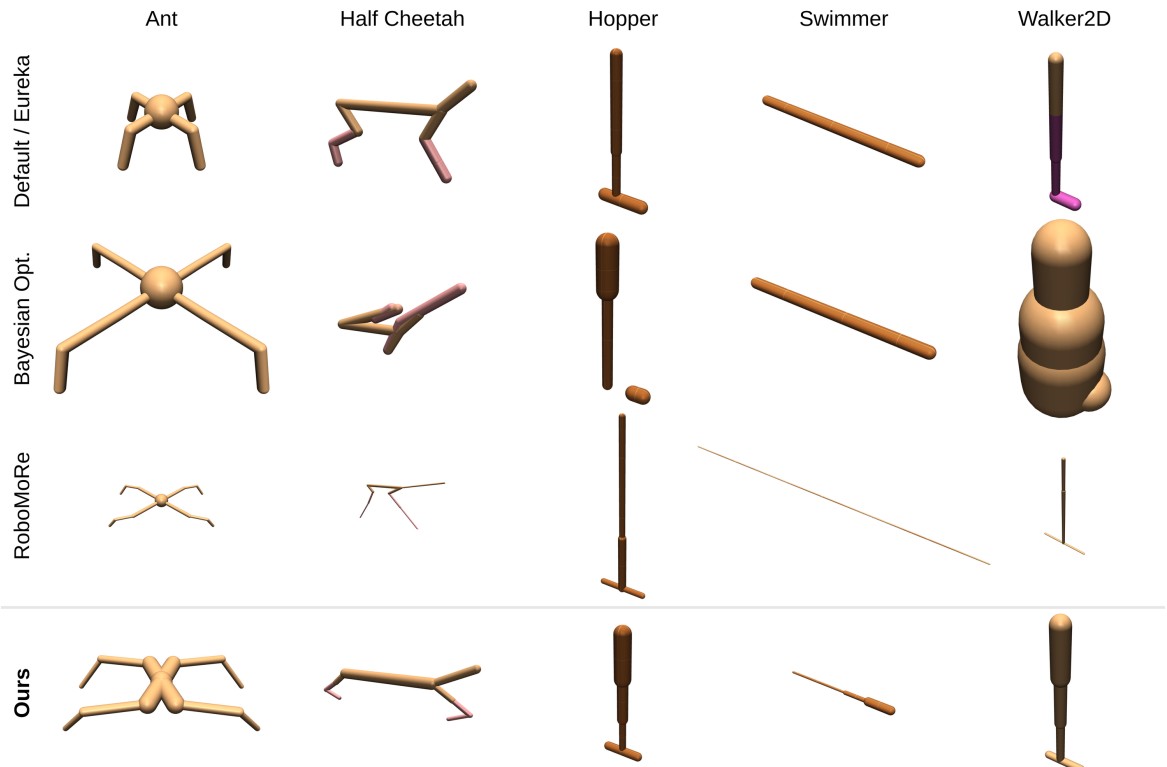

*Figure 4.* Representative final morphologies across tasks. D2C discovers non-trivial modifications (longer legs and wider stance on Ant, extended torso on HalfCheetah, asymmetric limbs on Walker2D) while baselines produce more conservative or invalid designs.

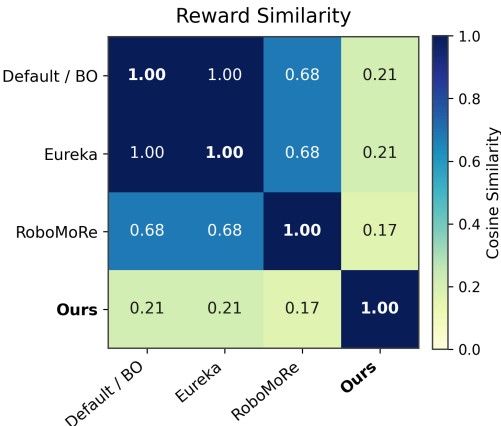

*Figure 5.* Reward-code similarity across methods (cosine similarity on AST-parsed feature vectors). Lower values indicate more distinct reward structures. Per-environment breakdown in Appendix D.

morphology alone underperforms the default body, yet the co-designed reward recovers a 50% gain. This is the failure mode that morphology-only or reward-only search cannot address cleanly. The role-separated debate gives the design agent a structured way to propose body changes, gives the control agent an explicit channel to flag controllability issues before training, and lets the archive retain empirical

winners when individual proposals fail.

**What the judges do and do not do.** The judge panel should not be interpreted as an LLM reward model. Judges do not select winners, assign scalar fitness, or replace simulation. They only translate measured behavior into textual critiques for the next round, while the archive is ranked by the external score $S$. This separation is important because it prevents persuasive but ungrounded LLM explanations from determining the final output. The evidence is consistent with judges acting as a directional search aid: speed and stability critiques predict leg and torso edits, and judge-conditioned runs make smaller, more targeted parameter changes than no-judge runs (Section 3.4; Appendix G.2). Thus the judges improve the proposal distribution rather than changing the objective.

**When the framework transfers.** Applying D2C to a new domain requires more than an LLM and a simulator: the domain must provide a constrained design parameterization, an executable reward template, and an external task score $S$ for archive selection. The third requirement is the most important. LLMs can propose reward code, but they should not define the metric that declares success. Under this contract, debate acts as a proposal mechanism for coupled design–reward artifacts, while simulator-measured

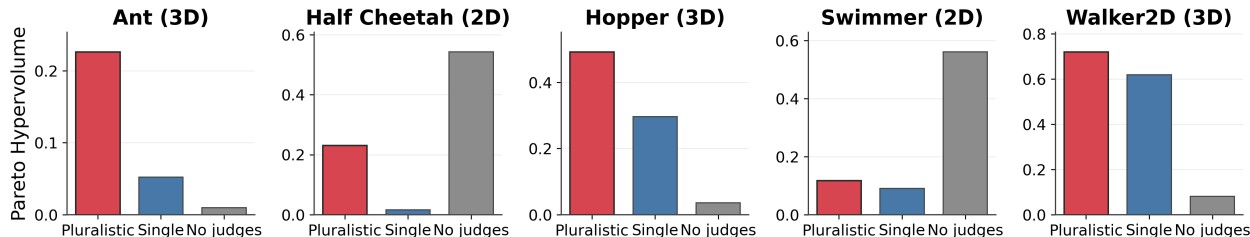

*Figure 6.* Pareto hypervolume over multi-objective metrics (3 search seeds, higher = broader coverage). Environments where stability is meaningful (Ant, Hopper, Walker2D) use 3D hypervolume (distance, stability, efficiency); others (HalfCheetah, Swimmer) use 2D (distance, efficiency). **Takeaway:** Criterion-specific judges improve coverage in environments with meaningful multi-dimensional trade-offs.

performance remains the arbiter. Domains without repeatable simulation or task-level success metrics would require a different evaluation protocol and should be treated as new experiments rather than direct transfers.

## 5. Related Work

**LLMs for robot design.** Using language models to assist robot design has emerged only recently (Qiu et al., 2026; Ringel et al., 2025). RoboMorph (Qiu et al., 2026) couples an LLM with evolutionary search to generate modular morphologies from language specifications. These pipelines typically optimize morphology under a fixed or hand-tuned objective and do not jointly reason about reward shaping for changed bodies. D2C targets design–reward co-design: morphology edits are paired with reward code and validated through simulator-in-the-loop training.

**LLMs for reward design.** A parallel thread of research explores using LLMs to automate reward design (Ma et al., 2024a; Kwon et al., 2023; Yu et al., 2023; Xie et al., 2024). For example, Eureka (Ma et al., 2024a) demonstrated that LLM-generated reward code can outperform hand-engineered rewards on challenging tasks by leveraging domain knowledge. Most existing approaches optimize reward code for a fixed morphology. D2C conditions reward generation on the proposed morphology and evaluates candidates in simulation, enabling reward shaping that co-adapts with morphological changes.

**Co-optimization of morphology and policy.** Given the strong coupling between a robot's body and its controller (Pfeifer et al., 2006; Sims, 1994; Cheney et al., 2018), prior work has studied joint optimization of morphology and control. Reward hacking and specification gaming motivate our separation of the learned reward $r$ (used for training) from the standardized evaluation score $S$ (used for selection) (Amodei et al., 2016; Krakovna et al., 2020). RoboMoRe (Fang et al., 2025) alternates between LLM-proposed morphology edits and reward hints in a coarse-to-fine fashion with a single LLM alternating roles. D2C differs in three ways: (1) it explicitly separates design and reward generation across two specialized agents, (2) it uses a dialectical thesis–antithesis–synthesis structure where agents critique each other's proposals before refinement, and (3) it employs criterion-specific judges that provide multi-objective feedback rather than relying on a single unified objective.

**Quality-diversity methods.** Quality-diversity (QD) algorithms such as MAP-Elites (Mouret & Clune, 2015) maintain archives of diverse, high-performing solutions across a behavior space (Pugh et al., 2016). These methods excel at discovering broad repertoires of designs but require hand-specified behavior descriptors and do not incorporate semantic reasoning about *why* certain designs succeed or fail. D2C has a different goal. Instead of filling a behavior archive, it seeks a single high-performing design through LLM-driven critique and refinement. The approaches are complementary—combining debate-generated candidates with QD archives is a promising direction we leave to future work.

**Multi-agent LLM debates.** Multi-agent debate has been studied as a mechanism to improve reasoning and evaluation (Irving et al., 2018; Liang et al., 2023; Du et al., 2024; Chan et al., 2023; Gu et al., 2024). For example, (Chan et al., 2023) show that debate-based protocols can increase evaluation reliability, and mixtures of judges have been used to reduce reward hacking in preference learning (Xu et al., 2024). D2C adapts this idea to robotics by grounding debate in a physics simulator. LLM judges condition on measured metrics and provide textual critiques, while selection is based solely on the task score $S$. Unlike prior debate frameworks that rely on LLM-learned judges or preference aggregation, D2C uses physics metrics as ground truth and LLM judges for qualitative guidance only. To our knowledge, D2C is the first to combine simulator-grounded multi-agent debate with criterion-specific LLM critics for morphology–reward co-design.

*Table 2.* Comparison with related LLM-based robotics methods. ✓ = directly optimized or used as a primary mechanism, ✗ = not a primary target. **Takeaway:** D2C combines morphology search, reward-code search, iterative refinement, and role-separated multi-agent debate.

| Method | Morph. | Reward | Iter. | Role sep. |
|---|---|---|---|---|
| RoboMorph | ✓ | ✗ | ✓ | ✗ |
| Eureka | ✗ | ✓ | ✓ | ✗ |
| Text2Robot | ✓ | ✗ | ✗ | ✗ |
| VLMgineer | ✓ | ✓ | ✓ | ✗ |
| RoboMoRe | ✓ | ✓ | ✓ | ✗ |
| **D2C (ours)** | ✓ | ✓ | ✓ | ✓ |

**Summary of key differences.** Table 2 summarizes the comparison axes most relevant to D2C. The closest methods are VLMgineer (Gao et al., 2025) and RoboMoRe (Fang et al., 2025), which also pair design changes with control objectives, but they do not use role-separated design–control debate with criterion-specific judges. D2C therefore differs in protocol rather than only in objective: specialized agents propose and critique design–reward pairs, while physics metrics ground both feedback and final archive selection.

## 6. Limitations and Future Work

**Compute and sample efficiency.** D2C is compute-intensive: each 5-round run requires 80 RL trainings per seed, totaling roughly 40 GPU-hours per environment in our setup. LLM inference is comparatively small (36 requests per debate round; Appendix C), while wall-clock time is dominated by simulator training. We reduce simulator load by skipping thesis training and parallelizing across GPUs, but scaling to larger design spaces will require more sample-efficient evaluation via early stopping, learned surrogates, or shared policy initialization.

**Simulator dependence and sim-to-real.** All experiments are conducted in simulation using Brax. Learned rewards and morphologies may exploit simulator artifacts and may not transfer to real robots without additional constraints. Extending D2C to sim-to-real settings will require robustness objectives (e.g., domain randomization), manufacturability constraints, and hardware-in-the-loop evaluation.

**Scope and design space.** We demonstrate D2C's effectiveness on five MuJoCo locomotion tasks using 3 search seeds and 5 final evaluation runs. Manipulation tasks and integration with quality-diversity methods (e.g., MAP-Elites) are natural extensions that could further characterize when debate-driven co-design is most effective. Our parametric design space (fixed XML template with editable parameters) ensures validity and stable training, although exploring topology changes with stronger geometry validators is a promising direction for future work.

**Proprietary LLM dependence.** Our main experiments

use GPT-5.2. Open-weight DeepSeek runs remain above the Default baseline on Ant and Swimmer (Appendix J), but do not match GPT-5.2. API behavior can also shift between model versions, so we report prompts and exact model identifiers for the main experiments.

**Benchmark familiarity.** MuJoCo locomotion tasks are widely documented, so LLM backbones may have seen related XMLs, reward code, or design heuristics during pre-training. This caveat tempers novelty and out-of-distribution claims, although the within-protocol co-design interaction does not depend on novelty alone.

**Single-metric archive selection.** The judge panel provides multi-objective feedback during exploration, but the archive selects candidates using a single scalar metric (cumulative distance from origin). The two operate at different stages: judges steer what to change next round, and the metric decides which candidate to keep. The ablation confirms that judges improve Pareto hypervolume even though selection is scalar (Figure 6). Reconciling multi-objective feedback with multi-objective selection (e.g., Pareto archives), and selecting judges per-environment based on preliminary evaluation (adaptive panel), are concrete directions for future work.

**Comparison scope.** We position D2C against LLM-based morphology and reward methods, including RoboMorph for morphology-only search, Eureka for reward-only search, and RoboMoRe for co-design, as well as black-box morphology optimization. The empirical baseline suite focuses on methods that can be evaluated under the same fixed-topology, per-candidate-RL protocol as D2C. The claim is therefore scoped to joint morphology–reward co-design under this protocol, not strict superiority over all robot co-design formulations.

## 7. Conclusion

We presented DEBATE2CREATE, a role-separated LLM debate framework for fixed-topology robot morphology–reward co-design. Across five MuJoCo locomotion tasks, D2C achieves the highest Default-normalized score among the evaluated baselines (up to $3.2\times$ on Ant and $\sim 9\times$ on Swimmer). The main empirical result is the co-design interaction: in 4/5 tasks, jointly optimizing morphology and reward outperforms either component alone, while learned rewards transfer to default morphologies in 4/5 tasks (Table 1). Within this protocol, the LLM agents have a bounded role: they propose constrained morphology–reward hypotheses, RL trains each candidate, and the fixed simulator score selects among them. This separation matters because a plausible XML or reward function is not evidence of a better robot; every candidate in D2C must survive the same training and evaluation protocol.

## Acknowledgments

This work was partially supported by the National Science Centre, Poland, under PRELUDIUM Grant No. 2024/53/N/ST6/03370 and Sonata Bis Grant No. 2024/54/E/ST6/00388.

## Impact Statement

DEBATE2CREATE is a research artifact for simulated robot co-design. It generates morphology parameters and reward functions, trains the resulting candidates, and selects them with a fixed simulator score. We intend it for research and prototyping, not direct deployment. The main risk is specification gaming: generated rewards can optimize proxies that fail under distribution shift. A fixed task metric $S$ reduces this risk but does not remove it. Designs or rewards produced by D2C should therefore not be used on hardware without sim-to-real validation, especially when robots may contact people. The method may also inherit biases from the LLMs used to propose designs and rewards. We did not test whether the method overlooks soft-bodied, asymmetric, or multi-modal robot classes, so extensions to new design spaces should include that check.

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

# Appendix

## Table of Contents

## A. Prompts

### A.1. Design Agent

Design-agent prompts are environment-specific. We show the Ant prompts below. Other environments follow the same structure with environment-specific parameter schemas and masked XML references.

**Base system prompt.**

```
## Mission
Design a robot morphology optimized for the task: {task_description}

## Environment
The ant is a 3D quadruped robot with a central torso and four legs. Each leg consists
    of three segments connected by hinge joints: an attachment segment from torso to
    hip, a thigh segment, and a shin/ankle segment. The robot moves forward by
    coordinating torque across eight actuated joints (two per leg).

## Parameters
Output **exactly 10 strictly positive** parameters that define the robot structure. Any
    zero or negative value will invalidate the design and be rejected:

**Torso:**
- `param1`: Torso radius (spherical torso size)

**Leg Attachment Offsets** (relative to torso center):
- `param2`: X-offset for leg attachment point
- `param3`: Y-offset for leg attachment point

**Thigh Segment Direction** (from attachment point):
- `param4`: X-offset for thigh segment endpoint
- `param5`: Y-offset for thigh segment endpoint

**Shin/Ankle Segment Direction** (from thigh endpoint):
- `param6`: X-offset for shin/ankle segment endpoint
- `param7`: Y-offset for shin/ankle segment endpoint

**Capsule Radii:**
- `param8`: Hip/attachment segment capsule radius
- `param9`: Thigh segment capsule radius
- `param10`: Shin/ankle segment capsule radius

## Constraints
- Every parameter must be strictly positive. Even slight negatives (e.g., `-0.01`) or
    zeros break the MuJoCo XML generation, so ensure all values are > 0 before
    returning them.
- The four legs use symmetric parameter values (front/back and left/right mirrored).
    Provide the parameter magnitudes once; the XML generator handles the sign flips
    internally.

Explore broadly: avoid tiny perturbations of prior designs and feel free to propose
    meaningfully different morphologies while respecting the positivity and symmetry
    rules.
```

**Base user prompt.**

```
## Task
Design a robot morphology that can achieve the following task: {task_description}

## Context
Use any provided debate context to spot issues, but rely on your own judgment to
    propose a distinct morphology rather than small tweaks.

## Masked XML Reference
```

```
Use the masked MuJoCo XML snippet below to understand how the parameters populate the
    model:
"""
<mujoco model="ant">
  <compiler .../> <option .../> <default .../> <asset .../>
  <worldbody>
    ...
    <body name="torso" pos="0 0 {height}">
      <geom name="torso_geom" pos="0 0 0" size="{param1}" type="sphere"/>
      <joint name="root" type="free" .../>
      <!-- Front-left leg (other 3 legs use mirrored signs) -->
      <body name="front_left_leg" pos="0 0 0">
        <geom fromto="0 0 0 {param2} {param3} 0" size="{param8}" type="capsule"/>
        <body name="aux_1" pos="{param2} {param3} 0">
          <joint name="hip_1" type="hinge" range="-40 40" .../>
          <geom fromto="0 0 0 {param4} {param5} 0" size="{param9}" type="capsule"/>
          <body pos="{param4} {param5} 0">
            <joint name="ankle_1" type="hinge" range="30 100" .../>
            <geom fromto="0 0 0 {param6} {param7} 0" size="{param10}" type="capsule"/>
          </body>
        </body>
      </body>
      <!-- front_right_leg, left_back_leg, right_back_leg: same structure with sign
    flips -->
      ...
    </body>
  </worldbody>
  <actuator>
    <motor joint="hip_1" .../> <motor joint="ankle_1" .../>
    ... <!-- 8 motors total: hip + ankle for each of 4 legs -->
  </actuator>
</mujoco>
"""

## Design Process
1. Analyze the task requirements and any provided context.
2. Apply your own reasoning to choose structural parameters; avoid tiny perturbations
    and aim for meaningful changes.
3. Provide a brief (<4 sentences) description explaining your design rationale and the
    expected gait/behavior.

## Output Format
Output strict JSON only:
{
  "parameters": [<param1>, <param2>, ..., <param10>],
  "description": "<your design rationale>"
}
```

**Thesis-phase additions.**

```
## Debate Context (Round {round_idx})
{feedback_context if available; else:
No prior best design -- explore broadly but keep proposals mechanically sound.}

## Response Checklist
- Propose parameters that make a structural change relative to any referenced design.
- Keep the `description` to <=2 sentences focusing on your rationale.
- Output strict JSON only.
```

**Synthesis-phase additions.**

```
## Thesis Design (JSON)
{thesis_json}
```

```
## Control Critique (Optional)
{critique_text}

## Synthesis Phase
Goal: propose an alternative morphology for the task: {task_description}
- Structural changes are encouraged; do not feel limited to small tweaks.
- Use the critique/reward context as optional guidance, but decide yourself what to
    change.
- Keep the `description` concise (<=2 sentences) explaining your rationale.

## Output Format (strict JSON only)
{
  "parameters": [<param1>, ..., <paramN>],
  "description": "short rationale"
}
```

## A.2. Control Agent

Control-agent prompts are shared across environments. At runtime, the system prompt is instantiated by substituting the reward signature, and the user prompt is filled with the environment observation code and task description. We include both the base prompt templates and the additional critique/reward requirements added during the debate loop.

**Base system prompt.**

```
## Role
You are a reward engineer trying to write reward functions to solve reinforcement
    learning tasks as effectively as possible.

## Goal
Write a reward function for the environment that will help the agent learn the task
    described in text.
Use useful variables from the environment as inputs.

## Signature
{task_reward_signature_string}

## Requirements
The function must be written in JAX and compatible with Brax.
- Use `jax.numpy` (aliased as `jp`) instead of NumPy or PyTorch.
- Do not use Torch tensors, TorchScript, or device-specific code.
- The function should return both the scalar reward and a dictionary of metrics.
- Keep the code pure and differentiable (avoid side effects or Python control flow that
    JAX cannot trace).
- Only reference modules that are already imported in the execution environment. You
    automatically have `import jax`, `import jax.numpy as jp`, and `Array = jax.Array`
    available -- do not introduce other undefined aliases (e.g., `JaxArray`) or
    external dependencies.
- If you use type annotations, rely only on these available symbols (e.g., `Array`) or
    standard typing constructs.
- Only access keys that exist in the provided `metrics` dict; do not invent metric keys.

## Checklist
- Keep the signature exactly as provided: {task_reward_signature_string}
- Use only jax/jp; no new imports or aliases
- Return reward and diagnostics dict; no extra outputs
- Do not introduce new function arguments or globals

## Output Format
Output only a single ```python``` fenced block and nothing else.
```

**Reward signature.**

```
def compute_reward(obs: jax.Array, action: jax.Array, prev_action: jax.Array, dt:
    float, metrics: Dict[str, jax.Array]) -> Tuple[jax.Array, Dict[str, jax.Array]]:
    """
    Returns:
      reward: scalar jax.Array (jit/vmap-safe)
      reward_components: dict of reward components (jax.Arrays), optional
    """
    ...
    return reward, reward_components
```

**Base user prompt.**

```
## Task
The Python environment is {task_obs_code_string}.

Write a reward function for the following task: {task_description}.

## Design Analysis
Before writing the reward function, consider:
1. **Design Characteristics**: What are the key features of this robot design?
2. **Control Challenges**: What control difficulties does this design present?
3. **Design Strengths**: What advantages does this design offer for the task?
4. **Design Weaknesses**: What limitations need to be addressed in the reward function?

## Reward Strategy
Based on the design analysis, create a reward function that:
- Exploits the design's strengths
- Compensates for the design's weaknesses
- Provides appropriate control signals for this specific design
- Guides the robot toward successful task completion
- Includes a leading docstring summarizing which design strengths/weaknesses the reward
    targets

## Learning From Previous Attempts
If feedback from previous rounds is provided, pay special attention to:
- What worked well in previous reward functions
- What didn't work and should be avoided
- Specific suggestions for improvement
- The actual code from previous reward functions (use as reference, don't copy)

Focus on creating a reward function that is specifically tailored to work well with
    this particular robot design, learning from previous attempts to improve
    performance.
```

**Output-format tip.**

```
## Output Format
The output of the reward function should consist of two items:
    (1) a scalar total reward (jax.Array),
    (2) a dictionary of reward_components {str: jax.Array}.
Output only a single '''python''' fenced block containing the function, and nothing
    else.

## Tips
  (1) Normalization / bounding:
      - Keep rewards numerically stable and within a reasonable range.
      - Prefer smooth saturations like jp.tanh(x), or carefully use jp.exp(-x) with a
    scale.
      - Clip with jp.clip to avoid NaNs/Infs (add small eps where needed).

  (2) Temperatures / scales:
      - If you transform a component (e.g., with jp.exp, jp.tanh), introduce a named
```

```
     temperature/scale
         parameter inside the function (e.g., temp_speed = 1.0). Do NOT add it as an
     input argument.
        – Each transformed component should have its own temperature/scale variable so it
     can be tuned independently.

   (3) Types and libraries:
        – Use JAX only: import jax.numpy as jp. Do NOT use PyTorch or NumPy.
        – Keep dtypes consistent (e.g., use jp.array(0.0, dtype=obs.dtype) for constants).
        – Avoid Python-side conversions like .item() that break jit/vmap.

   (4) Inputs and purity:
        – Only use the function arguments (obs, action, prev_action, dt, metrics). Do NOT
     reference self.* or globals.
        – Only access keys that exist in the provided `metrics` dict; do not invent
     metric keys.
        – Do NOT introduce new function parameters. Define any constants/weights inside
     the function.
        – The function must be pure and jit/vmap-safe: no randomness, no printing, no
     mutation of Python lists, and no control flow on array values (use jp.where).

   (5) Common shaping patterns:
        – Forward progress: prefer metrics["forward_reward"] if provided; otherwise fall
     back to the appropriate index in obs.
        – Energy/torque penalty: jp.mean(action**2)
        – Smoothness penalty: jp.mean((action – prev_action)**2) when prev_action is
     available.
        – Uprightness/stability: bounded terms via jp.tanh or distances with small eps
     (e.g., jp.sqrt(x*x + 1e-6)).
## Return Contract
  – Return either:
      reward
    or:
      reward, reward_components
  – The second form is preferred to enable logging of components.
```

### Critique-only pass additions.

```
System prompt addition:
Your sole job in this pass is to critique the morphology -- no code.

User prompt additions:
## Critique Requirements
- Provide a 'Critique:' section describing concrete weaknesses in stability, energy
    efficiency, fragility, and controllability.
- Prioritize failure modes that a reward function could expose.
- Keep the entire critique to ONE concise sentence because the text is saved verbatim.
- Do NOT provide code in this step.
```

### Reward-generation pass additions.

```
System prompt addition:
Write a reward function that drives fast, stable forward locomotion. Use the context as
    optional guidance; choose your own shaping terms.

User prompt additions:
## Optional Context (Critique Summary)
{critique_summary}

## Reward Requirements
1) Keep the docstring concise (<=2 sentences) stating the intent.
2) Use only the provided observations/metrics; do not invent new keys.
```

```
3) Output only Python code in a single '''python''' fenced block; no prose outside the
    fence.
```

**Execution-error feedback.**

```
## Error
Executing the reward function code above produced this exact error:

{traceback_msg}

## Instructions
Carefully analyze your previous answer and explain (in 2-3 sentences) why it failed,
    citing the traceback above. After you have articulated the root cause, briefly
    describe the specific adjustments you will make. Then provide a revised reward
    function that implements those adjustments. Keep the same JAX/Brax constraints
    (only `jax`, `jp`, and `Array = jax.Array` are available) and continue to return
    both the reward scalar and diagnostics dictionary.

Do NOT change the function signature or add/remove parameters. Do NOT introduce new
    imports or globals. Do NOT reference metric keys that were not provided; only
    access keys that exist in the `metrics` dict.

## Output Format
Keep code within a single '''python''' fence.
```

## A.3. Judges

Judge prompts are persona-conditioned and return a single-sentence observation as JSON. The personas are listed below.

```
Speed:
  System: You are a Performance-focused engineering evaluator. You optimize for forward
    distance/velocity and ensure morphologies are actually task-capable. You value
    designs that achieve maximum performance while maintaining task completion
    capability.
  Focus: forward distance, velocity, task capability, performance metrics, distance
    achieved

Stability:
  System: You are a Robustness-focused engineering evaluator. You look at falls,
    contact penalties, and upright posture. You ensure designs don't collapse or
    overfit to 'fast but fragile' solutions. You value stable, well-controlled designs.
  Focus: falls, contact penalties, upright posture, stability, robustness, control
    quality

Efficiency:
  System: You are an Energy/Control cost evaluator. You minimize control effort and
    torque use. You prevent solutions that 'burn energy' just to maximize distance. You
    value designs that achieve good performance with minimal energy expenditure.
  Focus: control effort, torque usage, energy efficiency, control cost, energy per
    distance

Novelty:
  System: You are an Exploration/Diversity evaluator. You score morphologies higher if
    they are structurally distinct from archive entries. You prevent collapse into
    repeated designs. You value structural diversity and morphological innovation.
  Focus: structural distinctness, morphological diversity, design uniqueness,
    preventing collapse
```

**Judge wrapper prompt.**

```
System prompt template:
## Role
```

```
{persona_system}

## Task
Evaluate the robot design. Focus on: {evaluation_focus}
Do not mention numeric scores or metric values. Provide a qualitative, mechanism-based
    observation only.

## Output Format
Return ONLY this JSON (no other text):
{
    "observation": "<ONE sentence: the single most important observation>"
}

## Rules
- Single sentence only; keep it under 30 words
- No lists, no bullet points, no elaboration
- State observations, not prescriptions
- If nothing notable, use null

User prompt template:
## Context
Round {round_index} | {persona_name} perspective

## Design Parameters
{design_parameters_json}

## Metrics
{training_metrics_json}

## Evaluation
{evaluation_metrics_json}

Return the JSON evaluation.
```

## B. Hyperparameters

Table 3 lists the RL training hyperparameters used for each environment. We select PPO or SAC based on which algorithm yields more stable training for each environment, then keep that choice fixed across methods and candidate morphologies within the environment.

Table 3. RL training hyperparameters for each environment.

| Hyperparameter | Ant | HalfCheetah | Hopper | Swimmer | Walker2D |
|---|---|---|---|---|---|
| algo | PPO | PPO | SAC | PPO | SAC |
| num_timesteps | 8,000,000 | 5,000,000 | 400,000 | 350,000 | 500,000 |
| episode_length | 450 | 400 | 800 | 600 | 1,000 |
| action_repeat | 1 | 1 | 1 | 1 | 1 |
| discounting | 0.97 | 0.95 | 0.997 | 0.97 | 0.997 |
| reward_scaling | 25 | 3 | 30 | 5 | 5 |
| learning_rate | 3e-4 | 3e-4 | 6e-4 | 3e-4 | 6e-4 |
| num_envs | 256 | 128 | 12 | 128 | 12 |
| batch_size | 512 | 512 | 48 | 128 | 48 |
| unroll_length | 5 | 15 | — | 6 | — |
| num_minibatches | 8 | 16 | — | 8 | — |
| num_updates_per_batch | 4 | 8 | — | 4 | — |
| entropy_cost | 0.01 | 0.001 | — | 0.001 | — |
| normalize_observations | — | — | true | — | true |
| min_replay_size | — | — | 2,048 | — | 4,096 |
| max_replay_size | — | — | 50,000 | — | 65,536 |
| grad_updates_per_step | — | — | 3 | — | 3 |
| max_devices_per_host | — | — | 1 | — | 1 |

## C. Evaluation Budget

**Candidate evaluations.** Under our default configuration (five debate rounds, $N_m$=4 morphology proposals per round, $N_r$=4 reward variants per morphology, and thesis designs used only for critique), each round evaluates $N_m \times N_r = 16$ synthesis design–reward pairs. Over 5 rounds, this yields $5 \times 16 = 80$ policy trainings per seed and environment. Candidate training budgets are fixed per environment (Table 3).

**LLM requests and token budget.** Each round performs (i) thesis generation for $N_m$ designs, (ii) critique for each thesis design, (iii) synthesis generation for $N_m$ designs, (iv) reward generation with $N_r$ samples per synthesis design, and (v) 4 judge-persona feedback calls on the top candidate only. With $N_m$=4 and $N_r$=4, this corresponds to 4 thesis generations, 4 thesis critiques, 4 synthesis generations, 4 synthesis critiques, $4 \times 4 = 16$ reward generations, and 4 judge calls, for a total of 36 LLM requests per round (assuming one request per sample). We estimate token usage from logged prompts and outputs across the three Ant runs (seeds 0–2) used in Figure 2; the resulting per-round totals are $\approx 1.40 \times 10^5$ input tokens and $\approx 3.54 \times 10^4$ output tokens.

*Table 4.* Approximate LLM request budget per debate round. Tokens/call are means over logged prompts/outputs; for batched calls, output tokens include all samples returned by that call.

| Component | Calls/round | Tokens/call (in/out) |
|---|---|---|
| Design (thesis) | 4 | 2.9k / 0.2k |
| Design (synthesis) | 4 | 3.2k / 0.1k |
| Control (critique) | 8 | 4.4k / 0.1k |
| Control (reward) | 16 | 5.1k / 1.0k |
| Judges (4 personas) | 4 | 0.3k / 0.05k |
| Total | 36 | — |

**Wall-clock time.** With 8 GPUs, we evaluate 8 design–reward pairs concurrently (one candidate per device). Since each round evaluates 16 candidates, we run two parallel waves per round. In our cluster setting, a debate round takes ~1 hour wall-clock, with wall-clock time dominated by RL evaluation.

## D. Reward Similarity and D2C Reward Functions

**Reward similarity metric.** We compute reward-code similarity by parsing each reward function into an AST and extracting a sparse, syntax-aware feature vector that captures (i) which normalized signals appear (e.g., forward velocity, health/uprightness, control cost), and (ii) the presence of common shaping motifs (e.g., smoothness penalties, contact terms, `tanh`/`exp`). Signals are canonicalized by renaming variables and collapsing equivalent aliases (e.g., `forward_reward`, `x_velocity`, `vx` ↦ `forward`; pitch/roll/yaw ↦ `orientation`). We then compute cosine similarity between feature vectors. Similarity is computed per environment and averaged across environments for Figure 5. This metric is intended as a coarse proxy for structural similarity of reward *code*, not functional equivalence.

**D2C reward functions (per environment).** We report the scalar reward expressions used for D2C in each task. Notation: $a$ is action, $a_{t-1}$ is previous action, $p$ is torso pitch, $\dot{p}$ is pitch rate, $u_z$ is the torso up-axis $z$ component, and $q_{\text{norm}}$ is the torso quaternion norm. All nonlinearities follow the implementations (e.g., `tanh`, `exp`, `clip`). These expressions are direct transcriptions of the rewards used during evaluation.

**Ant.**

$$r = 2.2 \tanh(v_x/3.0) + 0.7 \exp(-4(1 - u_z)^2) + 0.5(0.5 + 0.5 \operatorname{clip}(u_z, 0, 1))$$
$$- 0.15 \tanh((v_y/1.5)^2) - 0.25 \tanh((\omega_x^2 + \omega_y^2 + \omega_z^2)/4^2)$$
$$- 0.12 \tanh(((z - 0.75)/0.20)^2) - 0.10 \tanh((v_z/1.5)^2)$$
$$- 0.06 \|a\|_2^2 - 0.05 \|a - a_{t-1}\|_2^2 - 0.05 (q_{\text{norm}} - 1)^2.$$

**HalfCheetah.**

$$r = r_{\text{speed}} + 0.4 \, r_{\text{upright}} + 0.25 \, r_{\text{pitch\_rate}}$$
$$- (0.05 \, e_{\text{ctrl}} + 0.02 \, e_{\text{smooth}} + 0.001 \, e_{\text{jvel}} + 0.001 \, e_{\text{jang}}),$$

where $r_{\text{speed}} = 10\tanh(\text{fwd} \cdot \exp(-(p/0.6)^2)/10)$, $r_{\text{upright}} = \exp(-(p/0.5)^2)$, $r_{\text{pitch\_rate}} = \exp(-(\dot{p}/2.0)^2)$, and $e_{\text{jang}} = \mathbb{E}[\tanh(|\theta|/1.5)^2]$.

**Hopper.**

$$r = 2.0\,s + 1.0\,s_{\text{target}} + 1.0\,h - 0.6\,b - 0.5\,t - 0.3\,j - 0.08\,c - 0.05\,s_m - 1.0\,b_{\text{back}},$$

with $s = \tanh(v_x/2.0)$, $s_{\text{target}} = \tanh((v_x - 3.0)/2.0)$, $h = \text{clip}((z - 0.7)/0.3, 0, 1) \cdot \text{clip}((0.2 - |\theta|)/0.2, 0, 1)$, $b = \tanh((v_z/1.5)^2)$, $t = \tanh((\omega_{\text{torso}}/6.0)^2)$, $j = \tanh(\mathbb{E}[(\omega_{\text{joint}}/10.0)^2])$, $c = \mathbb{E}[a^2]$, $s_m = \mathbb{E}[(a - a_{t-1})^2]$, and $b_{\text{back}} = \tanh(\max(-v_x, 0)/1.0)$.

**Swimmer.**

$$r = 2.2\,r_{\text{fwd}} - 0.35\,p_{\text{slip}} - 0.22\,p_{\text{ang}} - 0.06\,p_{\text{act}} - 0.10\,p_{\text{smooth}} - 0.25\,p_{\text{back}},$$

where $r_{\text{fwd}} = \tanh(\text{fwd}/1.5)$, $p_{\text{slip}} = \tanh\left(\frac{|v_y|}{|v_x|+0.5}/0.6\right)$, $p_{\text{ang}} = \tanh\left(\frac{0.6\omega_0^2 + 1.0\omega_1^2 + 1.2\omega_2^2}{8}\right)$, $p_{\text{act}} = \tanh(\mathbb{E}[a^2]/1.0)$, $p_{\text{smooth}} = \tanh(\mathbb{E}[(a - a_{t-1})^2]/0.5)$, and $p_{\text{back}} = \tanh(\max(-\text{fwd}, 0)/0.5)$.

**Walker2D.**

$$r = 1.0\,\text{progress} + 0.15\,u\,h - 0.004\,c - 0.0025\,s_m - 0.02\,p_r - 0.02\,b - 0.0015\,f,$$

with $\text{progress} = v_{\text{fwd}} \cdot \tanh(0.6\,u + 0.4\,h)$, $u = \exp(-(p/0.6)^2)$, $h = \exp(-((z - 1.15)/0.35)^2)$, $c = \mathbb{E}[a^2]$, $s_m = \mathbb{E}[(a - a_{t-1})^2]/dt$, $p_r = (\dot{p}/3.0)^2$, $b = (v_z/1.5)^2$, and $f = \mathbb{E}[\omega_{\text{foot}}^2]$.

## E. Baseline Details

**Evaluation protocol.** All methods use identical RL training budgets per candidate and the same evaluation metric $S$. Main baseline comparisons report the best candidate found in each search seed, normalized by the Default score from that seed. The design–reward cross-over ablation retrains selected pairs 5 times to estimate policy variance. Scores are normalized by the Default baseline.

*Table 5.* Candidate budgets per method. All methods use the same RL training budget per candidate; differences are in what each method optimizes.

| Method | Candidates/seed | Morphology | Reward |
|---|---|---|---|
| **D2C (Ours)** | 80 | ✓ | ✓ |
| Eureka | 80 | ✗ | ✓ |
| RoboMoRe | 125 | ✓ | ✓ |
| Bayesian Opt. | 80 | ✓ | ✗ |

**RoboMoRe.** RoboMoRe (Fang et al., 2025) proposes morphology edits and reward modifications in a coarse-to-fine loop. For comparability, we evaluate RoboMoRe outputs inside our Brax stack and account for its search budget using the published coarse-stage candidate count of 25 morphologies $\times$ 5 rewards $= 125$ design–reward candidates per seed. Candidates are trained with the same RL budgets used for D2C and compared under the same evaluation metric $S$.

**Bayesian Optimization.** We include a black-box morphology-search baseline (Snoek et al., 2012; Shahriari et al., 2016) using the `scikit-optimize` library with a Gaussian process surrogate (Matérn-5/2 kernel), expected improvement (EI) acquisition, and 10 random initializations followed by 70 BO iterations (80 total evaluations, matching D2C). We run BO independently per environment and per PPO seed, and report the best-observed morphology in each BO trajectory (not the final iterate). The search space uses the same parameter bounds as D2C (Table 6) and invalid proposals (e.g., negative radii, self-intersecting geometry) are assigned score $S{=}0$ and included in the GP update. Approximately 12% of BO proposals were invalid across all runs. We evaluate resulting morphologies inside our Brax stack under the same default reward, RL budgets, and evaluation metric $S$ used for all methods. BO optimizes morphology only; we do not include reward search because reward functions are discrete code objects unsuitable for GP-based optimization.

**Baseline zero-score cases.** BO achieves $S{=}0$ on Hopper and Walker2D across all seeds, and RoboMoRe achieves $S{=}0$ on Walker2D. These are not pipeline bugs: we verified that (1) training runs complete without error, (2) morphologies are

valid XML, and (3) policies are saved and evaluated correctly. Under our protocol, the failures occur because these methods produce morphologies that appear kinematically unstable under the evaluation metric—the robot falls immediately or fails to make forward progress. BO's black-box search lacks semantic understanding of balance constraints, leading to proposals that satisfy parameter bounds but produce non-locomoting designs. RoboMoRe's coarse-to-fine search similarly converges to low-scoring morphologies on Walker2D. These results highlight the difficulty of co-design without structured critique mechanisms that identify stability issues early.

**Eureka.** We follow the published evolutionary search: 5 iterations per run and 16 reward samples per iteration (80 reward candidates per run), executed over 5 independent search runs. Each iteration performs in-context reward mutation using the best reward from the previous iteration and its reflection prompt. All rewards are evaluated on the default morphology with the same RL budgets as D2C. Eureka optimizes reward only; morphology is fixed to the default. Eureka uses 5 search runs compared to D2C's 3; as with the other $n=3$ comparisons, we treat differences as descriptive under the same evaluation metric.

**Default baseline.** Original morphology and hand-crafted reward from each MuJoCo task.

## F. Environment Details

For each MuJoCo locomotion task, we expose a subset of XML parameters for the design agent to modify (e.g., limb length/size and joint placement within bounded ranges) while keeping the overall kinematic structure fixed. All candidate XMLs are validated by the simulator before training. The evaluation score $S$ is the cumulative distance-from-origin used throughout the paper (Section 3.1), and all comparisons use identical RL training budgets per task (Table 3).

**Editable design parameterization.** For transparency, we summarize the per-environment parameterization exposed to the design agent:

*Table 6.* Design-space parameterization exposed to the design agent per environment. "Endpoints" refer to the parametric geometry values used to fill MuJoCo XML templates; the kinematic graph/topology is fixed.

| Environment | # Params | Exposed parameters (summary) |
| --- | --- | --- |
| Ant | 10 | Torso radius; leg attachment offsets; thigh/shin directions; capsule radii (symmetry enforced; all $> 0$). |
| HalfCheetah | 24 | Link endpoints (torso/head; rear/front leg chains) and capsule radii (2D; ensure no ground intersection). |
| Hopper | 10 | Vertical endpoints (torso/thigh/leg/foot; monotone in $z$); foot span ($x$); segment radii. |
| Swimmer | 6 | Segment lengths and radii for a 3-link chain (all $> 0$). |
| Walker2D | 10 | Vertical endpoints (torso/thigh/leg/foot; monotone in $z$); foot span ($x$); segment radii (symmetric legs). |

## G. Additional Results

### G.1. Statistical Significance Tests

Table 7 reports statistical comparisons between D2C and each baseline.

**Statistical test procedure.** For each environment and baseline, we compare the distribution of Default-normalized per-seed scores across $n=3$ independent seeds using a two-sided Welch's $t$-test (unequal variance). We report raw $p$-values and adjust for multiple comparisons over 15 tests (5 environments $\times$ 3 baselines) using Benjamini-Hochberg FDR correction at $\alpha = 0.05$. Given the small $n$, we treat these tests as descriptive and do not base claims solely on $p$-values; for some comparisons (e.g., BO/RoboMoRe with zero variance on Walker2D), the $t$-test is technically undefined but included for completeness.

**Effect sizes.** With $n=3$, effect sizes (Hedges' $g$) are more informative than $p$-values. Large effect sizes ($g > 2$) indicate that per-seed score differences are consistent across seeds, even when formal significance is underpowered. We treat effect sizes as the primary summary and encourage readers to inspect the per-seed scores in Table 8.

**Per-seed normalized scores.** For full transparency, Table 8 reports the Default-normalized scores for each method on each seed where the raw seed traces are included in this appendix.

*Table 7.* Exploratory statistical comparisons for main results ($n$=3 search seeds). We report two-sided Welch's $t$-test $p$-values and Hedges' $g$ effect sizes comparing D2C against each baseline per environment. Significance marker $^\dagger$ indicates $p < 0.05$ under Benjamini-Hochberg FDR correction, but these tests are descriptive at this sample size. **Takeaway:** Large effect sizes appear in many comparisons; per-seed values should be interpreted alongside these exploratory tests.

| Environment | Comparison | D2C Score | Baseline Score | $p$-value | Hedges' $g$ |
| --- | --- | --- | --- | --- | --- |
| Ant | Eureka | 3.18 | 0.94 | $0.0014^\dagger$ | 6.12 |
| | Bayesian | 3.18 | 2.54 | 0.0655 | 1.75 |
| | RoboMoRe | 3.18 | 0.87 | $0.0055^\dagger$ | 7.38 |
| HalfCheetah | Eureka | 1.36 | 1.00 | 0.1265 | 1.66 |
| | Bayesian | 1.36 | 0.66 | $0.0315^\dagger$ | 3.12 |
| | RoboMoRe | 1.36 | 0.14 | $0.0126^\dagger$ | 5.59 |
| Hopper | Eureka | 1.51 | 0.85 | $0.0074^\dagger$ | 3.66 |
| | Bayesian | 1.51 | 0.00 | $0.0042^\dagger$ | 10.02 |
| | RoboMoRe | 1.51 | 0.05 | $0.0041^\dagger$ | 9.63 |
| Swimmer | Eureka | 9.35 | 1.00 | $0.0011^\dagger$ | 20.04 |
| | Bayesian | 9.35 | 0.62 | $0.0006^\dagger$ | 20.38 |
| | RoboMoRe | 9.35 | 1.08 | $0.0007^\dagger$ | 19.48 |
| Walker2D | Eureka | 1.36 | 0.68 | $0.0142^\dagger$ | 3.41 |
| | Bayesian | 1.36 | 0.00 | $0.0073^\dagger$ | 7.58 |
| | RoboMoRe | 1.36 | 0.00 | $0.0074^\dagger$ | 7.57 |

*Table 8.* Per-seed normalized scores from search phase (method score / Default score, same seed). Each seed identifies its own best candidate; mean $\pm$ std computed over the 3 search seeds. Figure 2 summarizes the matched search-seed comparison, while Table 1 reports 5-run retraining for the design–reward cross-over study.

| Method | Seed | Ant | HalfCheetah | Hopper | Swimmer | Walker2D |
| --- | --- | --- | --- | --- | --- | --- |
| D2C | 0 | 3.08 | 1.21 | 1.56 | 9.28 | 1.18 |
| | 1 | 3.42 | 1.55 | 1.40 | 9.47 | 1.22 |
| | 2 | 3.03 | 1.31 | 1.56 | 9.29 | 1.69 |
| | *Mean $\pm$ std* | *3.18 $\pm$ 0.21* | *1.36 $\pm$ 0.18* | *1.51 $\pm$ 0.09* | *9.35 $\pm$ 0.11* | *1.36 $\pm$ 0.29* |
| Eureka | 0 | 0.91 | 0.96 | 0.88 | 1.02 | 0.74 |
| | 1 | 0.95 | 1.04 | 0.79 | 0.97 | 0.58 |
| | 2 | 0.96 | 1.01 | 0.87 | 1.00 | 0.73 |
| | *Mean $\pm$ std* | *0.94 $\pm$ 0.03* | *1.00 $\pm$ 0.04* | *0.85 $\pm$ 0.05* | *1.00 $\pm$ 0.03* | *0.68 $\pm$ 0.09* |
| BO | 0 | 2.71 | 0.58 | 0.00 | 0.55 | 0.00 |
| | 1 | 2.18 | 0.84 | 0.00 | 0.72 | 0.00 |
| | 2 | 2.72 | 0.57 | 0.00 | 0.60 | 0.00 |
| | *Mean $\pm$ std* | *2.54 $\pm$ 0.31* | *0.66 $\pm$ 0.15* | *0.00 $\pm$ 0.00* | *0.62 $\pm$ 0.09* | *0.00 $\pm$ 0.00* |
| RoboMoRe | 0 | 0.82 | 0.11 | 0.04 | 1.12 | 0.00 |
| | 1 | 0.95 | 0.18 | 0.07 | 1.01 | 0.00 |
| | 2 | 0.84 | 0.14 | 0.03 | 1.10 | 0.00 |
| | *Mean $\pm$ std* | *0.87 $\pm$ 0.07* | *0.14 $\pm$ 0.04* | *0.05 $\pm$ 0.02* | *1.08 $\pm$ 0.06* | *0.00 $\pm$ 0.00* |

## G.2. Critique-Action Correlation Analysis

To understand how criterion-specific judge feedback influences the design agent's decisions, we analyzed the relationship between critique themes and subsequent design parameter changes across all D2C runs.

**Methodology.** We extracted judge critiques from each debate round and categorized them by the issuing judge persona (Speed, Stability, Efficiency, Novelty). Design parameters were grouped into three morphological categories based on their functional role: *torso-related* (body/torso dimensions), *leg geometry* (limb lengths and positions), and *capsule sizes* (segment radii/thicknesses). We then tested whether critique themes co-occurred with changes to semantically-related parameter categories more often than expected by chance using chi-squared tests.

**Results.** Table 9 and Figure 7 summarize the co-occurrence analysis across 15 D2C runs (3 per environment; 300 total critiques, 60 design changes). All tested critique-parameter associations showed significant deviation from chance expectation ($p < 0.001$):

- **Speed (Legs)**: When the Speed judge provided feedback, leg geometry parameters changed in 140 instances (expected: 50; $\chi^2 = 162.0$).
- **Stability (Legs)**: Stability critiques co-occurred with leg changes 101 times ($\chi^2 = 52.0$).
- **Stability (Torso)**: Stability feedback also associated with torso modifications 87 times ($\chi^2 = 27.4$).
- **Efficiency (Size)**: Efficiency critiques co-occurred with capsule size changes 21 times, significantly *below* expectation ($\chi^2 = 16.8$), suggesting the design agent prioritizes other parameters when addressing efficiency concerns.

*Table 9.* Chi-squared test results for critique-parameter co-occurrences. All pairs significantly deviate from the expected count under independence ($p < 0.001$).

| Critique Type | Parameter Category | Observed | Expected | $\chi^2$ |
|---|---|---|---|---|
| Speed | Leg Geometry | 140 | 50 | 162.0 |
| Stability | Leg Geometry | 101 | 50 | 52.0 |
| Stability | Torso | 87 | 50 | 27.4 |
| Efficiency | Capsule Size | 21 | 50 | 16.8 |

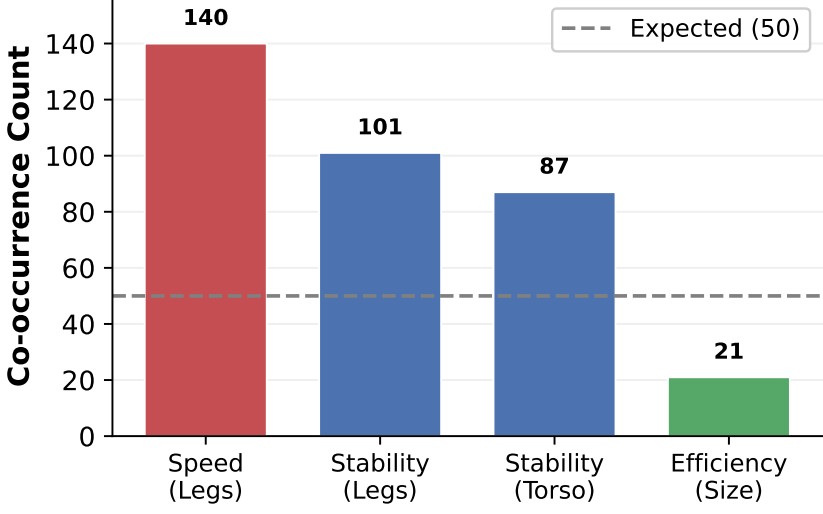

*Figure 7.* Co-occurrence counts between judge critique types and design parameter changes. Dashed line indicates expected count under independence. All deviations are statistically significant ($p < 0.001$, $\chi^2$ test).

Additionally, we compared design change magnitudes between D2C runs and control runs without criterion-specific judges using Mann-Whitney U tests. D2C runs exhibited significantly smaller total change magnitudes ($0.126 \pm 0.102$ vs. $0.220 \pm 0.134$; $p = 0.001$) and leg geometry changes ($0.089 \pm 0.079$ vs. $0.172 \pm 0.124$; $p < 0.001$), suggesting that multi-objective feedback leads to more targeted, incremental refinements rather than large exploratory jumps.

**Interpretation.**    These results provide evidence that the design agent responds to judge feedback in a topically coherent manner. Speed and stability critiques lead to locomotion-relevant parameter modifications, while efficiency concerns prompt different design strategies. The reduced change magnitudes in D2C runs suggest that criterion-specific judges help guide more focused exploration of the design space.

**Caveat (missing objectives).**    In environments where a judge's target metric is not logged or is near-constant (e.g., stability for HalfCheetah and Swimmer in our current setup), that judge's critique may be less grounded, and its inclusion can dilute the usefulness of feedback for multi-objective search. This aligns with the 2D hypervolume behavior discussed in Section 3.4.

### G.3. Example Debate Transcript

To illustrate the dialectical debate process, we present an example from an Ant run (seed 0 from the main experiments). This transcript shows rounds 0–2, where the score improved from 27,784 to 39,736 (+43%).

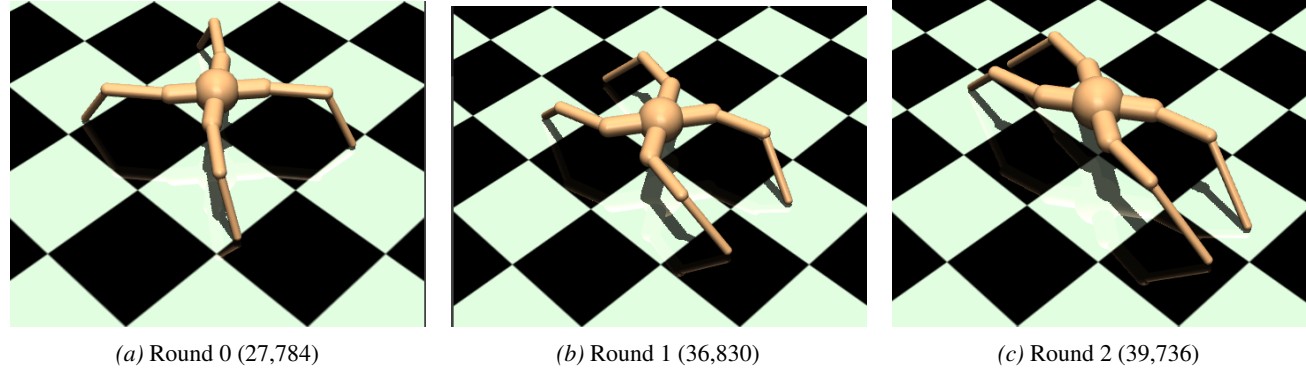

*(a)* Round 0 (27,784)          *(b)* Round 1 (36,830)          *(c)* Round 2 (39,736)

*Figure 8.* Morphological progression over debate rounds (Ant, seed 0). Key changes: torso radius increases (0.14→0.18), stance widens (0.22→0.30), shin extends (0.34→0.62). Score improves 43% from round 0 to round 2.

**Round 0: Initial Synthesis.**    The design agent proposes an initial morphology:

```
A smaller torso with wider, more lateral leg attachments increases roll stability at
    speed, while thighs and shins are angled more downward than forward to reduce
    scuffing and allow higher step frequency. Slightly thicker proximal links keep
    robustness for ground impacts, with light distal links to minimize swing inertia
    for a fast trot.
```

**Parameters:** torso=0.14, leg attachment=(0.22, 0.18), thigh=(0.24, 0.30), shin=(0.20, 0.34), capsule radii=(0.06, 0.04, 0.025).

**Round 0: Control Agent Critique.**    The control agent identifies potential failure modes:

```
The smaller torso and light distal links may encourage very fast but bounce-prone gaits
    that amplify pitch/roll oscillations and foot scuffing, making the system fragile
    to impacts and contact-force spikes, prone to wasted energy from high-frequency
    action jitter, and harder to control at speed without explicit rewards for
    uprightness, smooth torques, adequate ground clearance, and moderated contact
    impulses.
```

**Round 0: Judge Panel Feedback.**    After simulation (**score: 27,784**), the criterion-specific judges provide feedback:

```
Speed: The morphology achieves sustained high forward translation, indicating an
    efficient gait that prioritizes speed while remaining stable enough to maintain
    continuous locomotion.
```

```
Stability: Despite strong travel distance, the design likely prioritizes speed over
    robustness, showing marginal upright stability with increased contact events and
    fall susceptibility under perturbations.

Efficiency: The very large travel distance over a moderate episode length suggests high
    average speed likely sustained by continuous torque, indicating potentially high
    energy-per-distance despite good task performance.

Novelty: The parameter pattern suggests a fairly balanced, near-symmetric morphology
    with only modest extremes, indicating limited structural distinctness.
```

**Round 1: Revised Design.**   Informed by the critique, the design agent proposes:

```
A slightly larger torso and thicker proximal links trade a bit of agility for much
    better pitch/roll damping at high speed, reducing bounce and contact spikes. Very
    long, mostly lateral thigh+shin segments act like compliant outriggers that enable
    an aggressive fast trot/bound with long effective stride while keeping distal mass
    low for quick swing.
```

**Key parameter changes:**

- Increased torso radius ($0.14 \rightarrow 0.16$) for better pitch/roll damping
- Wider leg attachment ($0.22 \rightarrow 0.30$) for stability
- Longer shin segment ($0.34 \rightarrow 0.48$) for increased stride length
- Thicker proximal capsule ($0.06 \rightarrow 0.07$) for impact robustness

**Score: 36,830** (+33% over round 0). The Speed judge notes: "The morphology sustains fast, stable forward locomotion and greatly outpaces baseline distance." The Stability judge adds: "The long episode duration and strong travel suggest sustained upright, stable locomotion with few destabilizing contacts."

**Round 2: Continued Improvement.**   The control agent critiques:

```
The current reward shaping risks over-optimizing raw forward velocity for this
    long-stride, low-distal-mass design, leading to pogo-stick vertical bounce,
    pitch/roll oscillations, high contact-force spikes, and action jitter that produce
    fragile high-speed gaits.
```

The design agent responds:

```
Large torso with a very wide stance improves high-speed pitch/roll stability so the
    policy can push harder without tipping. Long, mostly lateral thigh plus longer shin
    increases effective stride length and ground contact leverage for fast
    trotting/bounding.
```

**Key parameter changes:**

- Increased torso radius ($0.16 \rightarrow 0.18$)
- Widened stance via Y-offset ($0.20 \rightarrow 0.30$)
- Further extended shin ($0.48 \rightarrow 0.62$)

**Final score: 39,736** (+8% over round 1, +43% over round 0). The Speed judge: "This morphology sustains very high forward travel with stable, efficient propulsion." The Stability judge: "Sustained upright locomotion with minimal instability is implied by the long episode duration and high reward."

**Summary.**   This transcript illustrates the core debate dynamics: (1) critiques identify concrete failure modes (bounce-prone gaits, pitch/roll oscillations); (2) the design agent responds with targeted parameter changes that address these concerns; (3) judge feedback validates improvements and surfaces new considerations; (4) iterative refinement yields a 43% improvement over two rounds. The archive mechanism preserves the best design from round 2 for the final output.

## H. Evaluation Metric Analysis

We use a cumulative distance-from-origin metric $S = \sum_{t=0}^{T-1} d_t$ rather than standard MuJoCo episode returns for evaluation. This section provides empirical evidence that $S$ correlates strongly with standard returns, validating our metric choice.

**Rationale.** Standard MuJoCo rewards include method-specific shaping terms (alive bonuses, control penalties, contact costs) that vary across reward designs. Using these returns for ranking would conflate task performance with reward engineering choices. Our metric $S$ is reward-agnostic: it measures only how far the robot travels, providing a fair comparison across methods with different reward structures.

**Correlation analysis.** Figure 9 shows the relationship between $S$ (cumulative distance) and standard episode returns. To ensure balanced comparison, we subsample 1,200 runs per environment from the candidate pool. Table 10 reports Spearman rank correlations, which are robust to non-linear relationships.

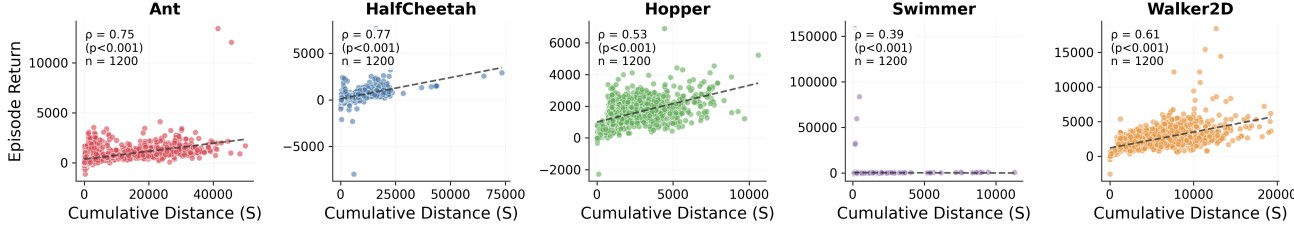

*Figure 9.* Correlation between cumulative distance metric $S$ and standard MuJoCo episode returns. Each point represents one training run. Dashed lines show linear regression fits. All correlations are statistically significant ($p < 0.001$).

*Table 10.* Correlation between cumulative distance $S$ and standard episode returns. Spearman $\rho$ measures rank correlation (robust to non-linearities). All correlations are statistically significant ($p < 0.001$).

| Environment | N | Pearson $r$ | Spearman $\rho$ |
|---|---|---|---|
| Ant | 1,200 | 0.53 | 0.75 |
| HalfCheetah | 1,200 | 0.58 | 0.77 |
| Hopper | 1,200 | 0.51 | 0.53 |
| Swimmer | 1,200 | −0.01 | 0.39 |
| Walker2D | 1,200 | 0.57 | 0.61 |
| **Mean** | **6,000** | — | **0.61** |

**Interpretation.** The mean Spearman correlation of $\rho = 0.61$ indicates that $S$ and standard returns share substantial rank agreement: designs that score well on $S$ generally also achieve high episode returns. The moderate (rather than perfect) correlation is expected: standard returns include method-specific shaping terms (alive bonuses, control penalties) that $S$ intentionally excludes to isolate locomotion performance. The weaker Pearson correlation for Swimmer ($r = -0.01$) reflects a non-linear relationship in that environment, but the positive Spearman correlation ($\rho = 0.39$) confirms that higher $S$ still corresponds to better-ranked performance. These results demonstrate that our metric choice does not systematically bias results away from standard benchmarks while providing a fairer comparison across methods with different reward structures.

**Ranking agreement for top candidates.** Table 11 provides additional evidence that selecting by $S$ does not sacrifice standard return performance. For each environment, we identify the best candidate by $S$ and report its percentile rank in terms of standard episode return. The best-$S$ candidate achieves a mean return percentile of 96.9%, indicating it typically ranks in the top 3% by standard returns. This confirms that $S$-based selection identifies high-quality designs under both metrics.

## I. LLM Backbone Sensitivity

To assess whether D2C's performance depends on LLM scale, we evaluated three OpenAI model variants on Ant: GPT-5-nano, GPT-5-mini, and GPT-5.2 (used in main experiments). Each configuration ran for 5 debate rounds with 3 seeds.

*Table 11.* Ranking agreement between $S$ and standard returns. "Return Percentile" indicates where the best-$S$ candidate ranks in terms of episode return (higher = better). The best-$S$ candidate is typically in the top 3% by standard returns.

| Environment | Return Percentile | Top-10 Overlap |
|---|---|---|
| Ant | 100.0% | 1/10 |
| HalfCheetah | 99.9% | 3/10 |
| Hopper | 99.9% | 1/10 |
| Swimmer | 98.2% | 0/10 |
| Walker2D | 86.6% | 0/10 |
| **Mean** | **96.9%** | 1.0/10 |

**Final performance.** Table 12 reports the best archived scores (normalized to the Default baseline). All three models achieve comparable final performance, with no statistically significant differences (pairwise t-tests, $p > 0.19$). This suggests D2C's gains stem from the debate framework structure rather than raw model capability.

*Table 12.* LLM backbone comparison on Ant (3 seeds $\times$ 80 candidates = 240 total per model). Normalized scores use the best archived result per seed. "Degenerate" counts candidates producing failed design–reward pairs (score $< 100$ or invalid). Differences in final normalized scores are not statistically significant ($p > 0.19$).

| Model | Norm. Score | Degenerate |
|---|---|---|
| GPT-5-nano | $3.60 \pm 0.26$ | 35.2% |
| GPT-5-mini | $3.23 \pm 0.32$ | 49.2% |
| GPT-5.2 (main) | $3.18 \pm 0.28$ | 15.8% |

**Robustness differences.** While final archived scores are comparable (due to the archive mechanism retaining the best candidate), smaller models produce substantially more degenerate candidates during the search process. GPT-5-mini has the highest failure rate (49.2% of candidates produce near-zero scores), followed by GPT-5-nano (35.2%). In contrast, GPT-5.2 achieves only 15.8% degenerate candidates. These failures represent wasted computation: each candidate requires a full RL training run before the score reveals the design was defective.

**Practical implications.** The archive mechanism makes D2C robust to individual candidate failures, allowing final performance to remain comparable across model scales. However, smaller models require evaluating more candidates to find viable designs, increasing compute costs. We used GPT-5.2 for main experiments as it provides the most reliable code generation with fewer degenerate outputs per round.

## J. Open-Weight Backbone Analysis

To assess whether D2C depends on a proprietary frontier model, we reran the full pipeline with two DeepSeek open-weight backbones on Ant and Swimmer. Each run uses the same protocol as the main experiments: 3 search seeds, 5 debate rounds, 80 trained candidates per seed, identical RL hyperparameters, and the same Default-normalized evaluation score. We compare against GPT-5.2, the backbone used in the main experiments.

*Table 13.* Open-weight backbone comparison under the same D2C protocol. Scores are Default-normalized best archived scores over 3 search seeds. DeepSeek-R1 retains 78% of GPT-5.2 performance on Ant and 71% on Swimmer; DeepSeek-V3 remains above the Default baseline on both tasks.

| Backbone | Access | Ant | Swimmer |
|---|---|---|---|
| GPT-5.2 | Proprietary | 3.18 | 9.35 |
| DeepSeek-R1 | Open-weight | 2.47 | 6.62 |
| DeepSeek-V3 | Open-weight | 1.85 | 3.72 |

The open-weight models do not match GPT-5.2, but both remain above the Default baseline on the two tested environments. The stronger DeepSeek-R1 result is consistent with D2C benefiting from explicit reasoning during critique and synthesis. The DeepSeek-V3 result is still useful: even without a reasoning trace, the debate loop produces viable design–reward pairs rather than collapsing to default-level performance. These experiments support the conclusion that D2C is not tied to a single proprietary backbone, although model capability affects sample efficiency and final score.

