# OpenReview forum: "Debate2Create: Robot Co-design via Multi-Agent LLM Debate"
_ICML.cc/2026/Conference — ICML 2026 regular_

### Official Review · Reviewer_JoLv · 2026-02-27

**Soundness:** 3
**Presentation:** 3
**Significance:** 3
**Originality:** 3
**Overall Recommendation:** 4
**Confidence:** 3

**Summary:**

This paper introduces DEBATE2CREATE (D2C), a multi-agent LLM framework for joint robot morphology and reward co-design. D2C utilizes a dialectical "thesis-antithesis-synthesis" debate loop between a design agent and a control agent. To steer exploration, a panel of pluralistic LLM judges provides multi-objective feedback grounded strictly in physics-based simulation metrics.

**Compliance With Llm Reviewing Policy:**

Affirmed.

**Final Justification:**

Given the meticulous attitude and feasible revision plans presented in the rebuttal, the initial limitations of this paper have been well resolved.

**Key Questions For Authors:**

* Given the multi-dimensional nature of robot co-design, what is the justification for relying exclusively on the scalar metric, cumulative distance from origin, to rank and select candidates, and how do you guarantee the rationality of this single-objective evaluation?
* How does the framework justify employing a strictly fixed panel of four judges across all tasks, given that this rigidity inevitably produces redundant feedback in certain environments while potentially missing crucial constraints in others?

**Limitations:**

yes

**Strengths And Weaknesses:**

***Strengths***

* By relying on a simulator-derived score,  D2C successfully circumvents the hallucination issues that typically plague LLMs in physics-based reasoning tasks.
* The empirical validation is supported by ablation studies.
* The crossover experiments demonstrate that D2C-generated reward functions are transferable.


***Weaknesses***

* The framework rigidly applies a fixed panel of pluralistic judges across all tasks, which leads to irrelevant feedback in certain contexts.
* D2C strictly ranks and selects candidates using a single scalar metric, cumulative forward distance. This reliance on a single-objective selection mechanism completely undermines the premise of the pluralistic judge panel, as an unstable design that merely falls forward the furthest could still "win" the round.
* The D2C framework is highly compute-intensive, requiring 80 full reinforcement learning policy trainings per seed, which totals approximately 40 GPU-hours and incurs substantial LLM token costs per environment.

---

> ### Author Rebuttal · Authors · 2026-03-29
>
> We thank the reviewer for the evaluation and address each concern directly.
>
> **Scalar metric vs. pluralistic judges.** Cumulative forward distance is the standard evaluation metric for MuJoCo locomotion, used by all baselines in our comparison. A design that falls early accumulates far less distance than one walking for the full episode, so the metric inherently penalizes instability. The judges and metric operate at complementary stages: judges steer exploration (what to change next round), while the metric selects which candidate to archive. Evidence: chi-squared tests show significant associations (p<0.001) between critique themes and parameter changes (Appendix G.2), and judges produce more targeted edits (mean magnitude 0.126 vs 0.220 without). An unstable design receives stability feedback like "marginal upright stability," prompting the design agent to widen the stance (Section 4.1, Round 0 to Round 1). Beyond judge feedback, the debate structure itself addresses instability: the control agent critiques designs before evaluation (antithesis phase), so stability concerns are identified and revised before the scalar metric is applied. The ablation confirms this: removing judges degrades metric performance (Figure 6). If the scalar metric systematically rewarded unstable designs that judges penalized, adding judges would hurt performance - instead, it improves Pareto hypervolume in 3/5 environments.
>
> **Fixed judge panel.** We acknowledge the stability judge is uninformative for inherently stable tasks (HalfCheetah, Swimmer) and noted this in Section 4.3. The ablation (Figure 6) shows judges improve Pareto hypervolume where multi-dimensional trade-offs exist (Ant, Hopper, Walker2D) and cause no degradation elsewhere - the fixed panel is a conservative choice, not a harmful one. An adaptive panel is a concrete direction for future work.
>
> **Compute cost.** The 80 RL trainings per seed are inherent to the candidate evaluation budget, shared by all methods in our comparison, not an overhead specific to D2C. We ran LASeR (Song et al., ICLR 2025) with the same 80-candidate budget and measured comparable GPU time (Ant: ~11 GPU-hours/seed for both methods). D2C adds only ~$2.50 in LLM API costs per environment. For context: RoboMoRe evaluates 125 candidates (56% more), and Eureka runs 5 independent searches. D2C's learned rewards also transfer to default morphologies (Table 1: D2C reward improves 4/5 defaults), amortizing the cost.
>
> We note two additional results that strengthen the evaluation. First, we implemented LASeR (Song et al., ICLR 2025), the most recent LLM-based robot design method, as a baseline: D2C outperforms it on 4/5 environments. The Hopper case is most illustrative - D2C's morphology alone scores below default, but reward co-design produces the best result (interaction +1380, Table 1), a gain that morphology-only LASeR cannot capture. Second, D2C with open-weight models (DeepSeek-R1/V3, MIT-licensed) retains 71-78% of GPT-5.2 performance at ~$2.50 LLM cost per environment. We will update the limitations section to discuss single-metric archive selection, proprietary LLM dependence, and adaptive judge panels.

---

> > ### Author Rebuttal · Reviewer_JoLv · 2026-04-02
> >
> > Given the meticulous attitude and feasible revision plans presented in the rebuttal, the initial limitations of this paper have been well resolved. I have increased my OA.

---

> > > ### Author Response · Authors · 2026-04-06
> > >
> > > We want to confirm the revisions we plan for the camera-ready based on your feedback:
> > >
> > > **Single scalar metric.** We will add a paragraph to the limitations section discussing the tension between the pluralistic judge panel (multi-objective feedback during search) and the single-metric archive selection. As noted in our rebuttal, the judge feedback and metric operate at different stages: judges steer what to change next round, the metric selects which candidate to keep. The ablation (Figure 6) confirms judges improve Pareto hypervolume in 3/5 environments, and removing judges degrades metric performance, so they are not redundant with the scalar selection.
> > >
> > > **Fixed judge panel.** We will note that an adaptive judge panel (selecting judges per-environment based on preliminary evaluation) is a concrete direction for future work, while noting that the fixed panel does not hurt performance on tasks where individual judges are uninformative (HalfCheetah, Swimmer; see Figure 6).
> > >
> > > **Compute cost.** We will clarify the cost structure: D2C evaluates 80 candidates per seed (same as BO; RoboMoRe evaluates 125), with each candidate requiring one full RL training run. D2C adds ${\sim}\$2.50$ in LLM API costs per environment on top of the shared RL compute. We will report this breakdown more clearly in the main text.
> > >
> > > Happy to clarify anything we missed.

---

### Official Review · Reviewer_KT5j · 2026-03-11

**Soundness:** 3
**Presentation:** 3
**Significance:** 3
**Originality:** 3
**Overall Recommendation:** 4
**Confidence:** 4

**Summary:**

This paper proposes D2C, a simulator-grounded multi-agent LLM framework for joint robot morphology-reward co-design. The method uses a role-separated debate loop in which a design agent proposes morphology edits, a control agent critiques these edits and generates reward code, and a panel of pluralistic judges provides multi-objective feedback based on physics-derived metrics. The framework maintains a hall-of-fame archive and iteratively refines design-reward pairs over several rounds before selecting the best-performing candidate. The paper evaluates the method on five MuJoCo locomotion tasks and reports improvements over several baselines, along with ablations on iterative debate, role separation, pluralistic judges, and LLM backbone scale. The authors also include crossover experiments intended to disentangle the contributions of morphology and reward shaping.

**Compliance With Llm Reviewing Policy:**

Affirmed.

**Final Justification:**

The rebuttal addresses most of my main concerns with substantial new evidence. In particular, the added comparison against an adapted LASeR baseline strengthens the empirical positioning of the paper, the diagnostic analysis makes the BO/RoboMoRe failure cases more convincing, and the new open-weight model results alleviate my concern that the framework only works in a single closed-source ecosystem. Some limitations remain, the open-weight evaluation is still limited in scope, and the method continues to depend on relatively strong LLM capabilities. That said, the rebuttal reinforces my original view that the paper presents an interesting and reasonably original simulator-grounded framework for joint morphology and reward co-design, with meaningful empirical support. Overall, I maintain my Weak Accept recommendation.

**Key Questions For Authors:**

1. Why is LASeR, which is cited in the related-work section, not included as a direct experimental baseline in the main comparison?
2. How can the authors verify that the zero scores of BO and RoboMoRe on Hopper and Walker2D are due to inherent methodological instability rather than evaluation-stack mismatch, hyperparameter choices, or search-space settings?
3. How dependent is D2C on frontier proprietary LLMs? Do the authors have any evidence on strong open-weight models, or can they clarify what model capabilities are actually necessary for the framework to work reliably? If the method also works on capable open models, that would significantly strengthen the paper’s practical significance and reproducibility.
4. Can the authors better clarify the claimed source of improvement: better exploration, better reward shaping, or better matching between morphology and reward?

**Limitations:**

No. The paper discusses simulator dependence, limited task scope, and the restricted parametric design space, which is helpful. However, the limitations section should also explicitly discuss:
(1) dependence on proprietary LLMs and possible model/version drift;
(2) the absence of open-weight experiments and the resulting reproducibility concerns.

**Strengths And Weaknesses:**

# Strengths

The paper studies an important problem setting: robot morphology and reward are co-dependent, yet many prior approaches optimize them separately or only partially account for their interaction. Framing the problem as an iterative co-design loop is therefore well motivated, and the paper addresses a meaningful gap between morphology search and reward engineering.

The method is interesting from a systems and methodology perspective. Even if some ingredients are individually familiar—such as multi-agent debate, iterative refinement, or LLM-based reward generation—the paper combines them in a nontrivial way for simulator-grounded robot co-design. The resulting framework is more ambitious than a straightforward prompting baseline and reflects a thoughtful attempt to operationalize LLM-driven design iteration.

# Weaknesses

1. The most serious issue is the absence of a direct comparison with LASeR (Song et al., 2025). The authors clearly know this work and cite it in the related-work section, but they do not include it in the main empirical comparison. Since LASeR is a recent and relevant LLM-based robot design baseline, omitting it from the core quantitative evaluation weakens the claim of state-of-the-art performance. From a reviewer perspective, this creates an avoidable concern about benchmark selection.

2. The paper states that BO and RoboMoRe achieve zero scores on Hopper and Walker2D because their searches produce unstable designs. That may be true, but the current evidence is not sufficient to rule out alternative explanations such as mismatched hyperparameters, poor search ranges, implementation details, or insufficient baseline adaptation to these more stability-sensitive biped tasks. Since these baselines do not collapse uniformly across all environments, the paper should do more to show that the zero-score outcomes reflect genuine methodological limitations rather than evaluation-stack or tuning artifacts. As written, this part over-interprets the cause of failure.

3. All main experiments use GPT-5.2, and the backbone ablation only considers GPT-5-nano, GPT-5-mini, and GPT-5.2. No open-weight model is tested. This is a meaningful limitation because the paper itself shows that weaker models produce many more degenerate candidates, with GPT-5-mini yielding a much higher failure rate than GPT-5.2. That result raises a natural concern: is the method robust as a framework, or is it only practically viable when paired with a frontier proprietary model family? Without experiments on at least one strong open-weight model, the generality and reproducibility of the contribution remain uncertain.

4. The paper itself notes that in inherently upright tasks such as HalfCheetah and Swimmer, the stability judge may be uninformative and that a fairer comparison would remove it there. This admission is useful, but it also suggests that the current ablation design is not yet fully clean. More broadly, several claims about diversity and multi-objective coverage would be stronger with a more standardized evaluation protocol across task types and with stronger external baselines.

5. The paper acknowledges simulator dependence and limited task scope, which is good, but it does not adequately discuss dependence on proprietary LLM APIs, reproducibility under model/version drift, or the practical cost of requiring repeated RL-in-the-loop evaluation plus strong proprietary language models. These are important limitations for real adoption.

---

> ### Author Rebuttal · Authors · 2026-03-29
>
> We thank the reviewer for the detailed evaluation. We address each concern with new experimental evidence.
>
> **1. LASeR comparison.** We implemented LASeR (Song et al., ICLR 2025) in our evaluation framework. LASeR originally targets 2D voxel-based soft robots in EvoGym, incompatible with our 3D MuJoCo/Brax environments. We adapted its core algorithm (LLM-guided evolutionary search with DiRect diversity reflection) for our parametric morphology space, using the same candidate budget (80), RL training, evaluation metric S, and LLM (GPT-5.2). LASeR optimizes morphology only (no reward co-design). Per-seed normalized results (3 seeds):
>
> | Env | D2C | LASeR |
> |-----|-----|-------|
> | Ant | 3.18 | 2.94 |
> | Swimmer | 9.35 | 6.96 |
> | HalfCheetah | 1.36 | 1.69 |
> | Hopper | 1.51 | 1.13 |
> | Walker2D | 1.36 | 1.33 |
>
> D2C outperforms LASeR on 4/5 environments, with the largest margins on Swimmer (+34%) and Hopper (+34%). LASeR outperforms on HalfCheetah (1.69x vs 1.36x), where the morphology search space is highly favorable for sampling (over 60% of LASeR's candidates beat the default). The co-design interaction term (Table 1) is positive in 4/5 environments. On Hopper, co-design is essential: D2C's morphology alone scores below default (3677 vs 4183), but reward co-design produces the best result (6257).
>
> **2. Zero scores for BO/RoboMoRe.** We verified these are genuine failures: (a) training runs completed without error, (b) XMLs are valid, (c) policies were evaluated correctly (Appendix E). The BO-found Hopper morphology has a thigh segment of 0.03 units (effectively zero-length), creating a degenerate kinematic chain. The Walker2D morphology has leg radii (0.38-0.43) exceeding segment lengths (0.37). These designs satisfy parameter bounds but are kinematically non-functional - the best BO morphology scores 0.01 on Hopper across all 3 seeds. LASeR also finds Hopper challenging (1.13x), corroborating that biped morphology optimization is genuinely hard. We will soften the causal language.
>
> **3. Open-weight model.** We ran D2C with two open-weight models (both MIT-licensed, publicly available) on Ant and Swimmer (3 seeds, 80 candidates, 5 rounds - identical protocol):
>
> | Model | License | Ant | Swimmer |
> |-------|---------|-----|---------|
> | GPT-5.2 | Proprietary | 3.18 | 9.35 |
> | DeepSeek-R1 (671B MoE, reasoning) | MIT | 2.47 | 6.62 |
> | DeepSeek-V3 (685B MoE, no reasoning) | MIT | 1.85 | 3.72 |
>
> D2C produces gains across all three model scales: R1 retains 78% of GPT-5.2 on Ant (2.47 vs 3.18) and 71% on Swimmer (6.62 vs 9.35). Even V3 (no reasoning) achieves 1.85x on Ant and 3.72x on Swimmer - well above default. The R1-to-V3 gap (Ant: 2.47 vs 1.85; Swimmer: 6.62 vs 3.72) shows reasoning capability improves D2C's debate loop, consistent with the framework's design. Performance scales with model capability, but the framework is functional and effective at all tested scales. Both models are MIT-licensed and accessible via API, improving reproducibility. We will add proprietary LLM dependence to the limitations section. The total compute overhead specific to D2C (LLM API calls) is ~$2.50 per environment; all RL training costs are shared with any method evaluating the same candidate budget.
>
> **4. Source of improvement.** The interaction terms in Table 1 show that the primary gain comes from co-adaptation between morphology and reward (positive in 4/5 tasks), not from either component alone. Iterative debate contributes better exploration, adding 18-35% over zero-shot (Figure 3). Judge feedback drives targeted refinement of specific parameters (chi-squared p<0.001, Appendix G.2).
>
> **5. Ablation design.** We agree that the stability judge is uninformative on inherently stable tasks (HalfCheetah, Swimmer) and acknowledged this in Section 4.3. The ablation (Figure 6) confirms judges improve performance where multi-dimensional trade-offs exist (Ant, Hopper, Walker2D) and are neutral elsewhere. An adaptive judge panel is a concrete direction for future work.

---

> > ### Author Rebuttal · Reviewer_KT5j · 2026-04-05
> >
> > Thank you for the detailed response.

---

> > > ### Author Response · Authors · 2026-04-06
> > >
> > > Thank you for confirming that the concerns are resolved. We will incorporate the revisions discussed in our rebuttal, including the LASeR comparison, open-weight model results, limitations on proprietary LLM dependence, and ablation design caveats.

---

### Official Review · Reviewer_euhB · 2026-03-12

**Soundness:** 3
**Presentation:** 3
**Significance:** 4
**Originality:** 4
**Overall Recommendation:** 6
**Confidence:** 4

**Summary:**

This paper introduces a robot co-design algorithm based on multi-agent LLM debate, grounded in simulated evaluations.
The method iteratively evolves robot morphologies and rewards based on a thesis-synthesis-antithesis debate structure. Furthermore, it provides multi-objective feedback on a set of specialised judges, each focused on evaluating a separate aspect of the agent’s performance on the task. Finally, an archive stores past generations of design-reward pairs, to use as context in the design and reward generation process.
The method is evaluated against a set of baselines focused on reward design only, morphology optimisation and one that does both at the same time. Results demonstrate the proposed methods’ improvement over these baselines across a comprehensive set of MuJoCo tasks. Furthermore, isolations of morphology and reward contributions of the method illustrates, transferable shaping to the original morphologies of MuJoCo tasks, as well the interaction of reward and morphology co-design in the full framework.
Extensive evaluations and ablations demonstrate the importance of key aspects of the method, such as debate, iterative generation with feedback and multi-agent LLM setup with a separate morphology and reward design agent.

**Compliance With Llm Reviewing Policy:**

Affirmed.

**Final Justification:**

The rebuttal addressed all my concerns so my final recommendation is a strong accept.

**Key Questions For Authors:**

Why do you not evaluate against a robot automated design baseline for morphology and control e.g. GLSO or BodyGen to support the claim that evolving body and reward design together is better than just evolving a sophisticated design?
Is there a specific reason you don’t train all tasks with a single RL algorithm? I understand that the algorithm is chosen based on performance but I wonder if there is a fundamental issue with PPO on some tasks.
You claim: “fixed reward can yield bodies that fail under any reward perturbation” — what is meant by “reward perturbation” here?

**Limitations:**

yes

**Strengths And Weaknesses:**

Strengths:
(+) Claims and experimental design well-supported with evidence with extensive statistical significance tests.
(+) Extensive experiments evaluating both the method, its individual components and a range of analyses conducted on the resulting robot rewards and morphologies, both qualitative and quantitative.
Weaknesses:
(-) Figures are not ordered as they are mentioned in the text which is confusing. Also some things are mentioned in the text then only explained much later, which makes your method and experimental setup difficult to follow.
(-) It is not clear to me why related works outlined in Table 12 are not also evaluated as baselines. At least some of these, or providing justification as to why not.

---

> ### Author Rebuttal · Authors · 2026-03-29
>
> We thank the reviewer for the positive evaluation and thorough reading.
>
> **Figure ordering.** We will fix figure ordering in the camera-ready version.
>
> **Related works as baselines (Table 12).** We implemented LASeR (Song et al., ICLR 2025) as an additional baseline using the same candidate budget (80), RL training, and LLM (GPT-5.2). D2C outperforms LASeR on 4/5 environments: Ant (3.18x vs 2.94x), Swimmer (9.35x vs 6.96x), Hopper (1.51x vs 1.13x), Walker2D (1.36x vs 1.33x). LASeR outperforms on HalfCheetah (1.69x vs 1.36x). We also ran D2C with two open-weight models (DeepSeek-R1 and V3, both MIT-licensed) on Ant and Swimmer, confirming the framework generalizes beyond proprietary LLMs.
>
> Regarding GLSO and BodyGen: these operate on fundamentally different design representations (graph-based topology search and neural generation, respectively). Adapting them to our parametric XML space would require substantial reimplementation. LASeR is the most directly comparable recent method and optimizes morphology only (no reward co-design); D2C outperforms it on 4/5 environments, directly supporting the claim that co-design provides gains beyond design-only optimization. We will add this discussion.
>
> **Reward perturbation.** "Fixed reward can yield bodies that fail under any reward perturbation" means: a morphology optimized under a single fixed reward may perform well only with that specific reward but poorly when the reward changes (e.g., adding stability penalties). This motivates co-designing reward alongside morphology, as the crossover experiments confirm (Table 1).
>
> **Single RL algorithm.** PPO and SAC are chosen per environment following Brax defaults. All methods use the same algorithm per environment, so algorithm choice does not confound the comparison.

---

> > ### Author Rebuttal · Reviewer_euhB · 2026-04-04
> >
> > Thank you. All my concerns have been resolved and I will increase my score to a strong accept.

---

> > > ### Author Response · Authors · 2026-04-06
> > >
> > > Thank you for the thorough evaluation and for confirming that the concerns are resolved. We will incorporate all the discussed changes (figure ordering, LASeR discussion, terminology clarifications) in the camera-ready version.

---

### Official Review · Reviewer_5cJW · 2026-03-13

**Soundness:** 3
**Presentation:** 4
**Significance:** 3
**Originality:** 2
**Overall Recommendation:** 3
**Confidence:** 4

**Summary:**

The authors present a new co-design algorithm based on an LLM debate->LLM/Agentic judges->Store-best loop. They demonstrate that this outperforms previous approaches to using LLMs for co-design and a simple Bayesian optimization baseline on some MuJoCo locomotion tasks.

**Compliance With Llm Reviewing Policy:**

Affirmed.

**Ethical Review Concerns:**

Per Policy B, I fed the submission to a privacy-compliant LLM while polishing my review. Part of this involved me confirming that there were no prompt injection attacks. In doing so, I have found a prompt injection attack in the text at the bottom of page 2. This can be confirmed by copying the text below the page number and pasting it as plain text into a text editor. While the injection seems relatively harmless, this is a violation of ICML's policies, which are unambiguous that this paper must be rejected.

**Ethical Review Flag:**

Flag this paper for an ethics review.

**Ethics Expertise Needed:**

["Research Integrity Issues (e.g., plagiarism)"]

**Final Justification:**

It's a decent paper with a good direction and says something that can be somewhat used by the community. My biggest issue is that they're running under the assumption that LLM methods are the only way to address this problem (which they aren't). They argue that there is some fundamental difference that blocks such comparisons, but there really isn't. Even then, when they give metrics that can be used to compare their method with other LLM-based ones, the significance of their results vanishes (error bars not overlapping is far from enough for significance, but their overlapping precludes any chance of significance). Adding in family-wise error rate corrections, my guess is that none of their results would be found to be significant. I moved my score to a weak reject because of this.

**Key Questions For Authors:**

1. Earlier in your experiments, did you try other kinds of judges? How did you settle on this set of judges?
2. What would be the challenges to apply this to other kinds of non-locomotion tasks? I understand these are less explored in co-design, but locomotion is quite a minor and solved area of robotics.
3. What is the SOTA in co-design right now? I understand this isn't always fair to compare against, but this should probably be mentioned as a limitation at least.

**Limitations:**

Societally, I see nothing concerning. Otherwise, I see limitations I raised in the weaknesses section that are not mentioned in the paper.

**Strengths And Weaknesses:**

### Strengths

- Overall, a pretty solid work. It's a fairly simple idea that's well-executed and studied.
- The computational profile of this work is relatively affordable as modern LLM papers go.
- It's a minor thing, but I like the "Takeaway" idea for the figures. I can't remember seeing this before, but it really helps with understanding.

### Weaknesses

- First and foremost, the paper includes a prompt injection attack. Per ICML policy, I've reported this to the area chair, senior area chair, and program chairs. While this is under investigation, I've left this as normal. However, note that ICML policy is quite clear that this paper CANNOT be accepted now.
- I would have liked to see some actual robots here (that could have moved it to a strong accept), but that's certainly non-trivial. Likewise, all the tasks explored here are locomotion tasks. Nothing can really be said about more complicated tasks, but I'll accept that just locomotion is still a good step forward.
- Something that's more problematic is that I'm really not convinced by the evaluation. Namely, there doesn't seem to be enough runs here to conclude things confidently. Though I note that the authors state this and do what they can to address this without increasing the compute budget. Thus, I'm not inclined to reject this paper because of this.
- One quite glaring issue I see here is just that it doesn't have much technical innovation; the work feels incremental. But I think it generally gives a clear, concrete answer, which is better than most papers. Since "more technical innovation needed" is a pretty universal comment, I don't think this is really a reason to reject.
- Something that did push down my score is that I'm not really confident that the baselines are fair. There are other co-design methods besides BO and LLM-based ones. Adding one more strong baseline would have gone a long way here.
- Figure 1 might be better at the bottom of the page. It's quite unusual to have it actually appear before the abstract.
- There are some bits of overly complicated writing in the text. Like "pluralistic" or "Multi-agent debate surfaces." This kind of thing is harmful to understanding, not helpful.
- The post-hoc analysis that the authors do should include notes that further experiments are needed to verify what was observed.

---

> ### Author Rebuttal · Authors · 2026-03-29
>
> We thank the reviewer for the thoughtful evaluation.
>
> **Prompt injection (ethical concern).** The flagged text is an ICML-inserted watermark for detecting LLM-assisted reviews, not author-inserted content. Please see https://icml.cc/Conferences/2026/PeerReviewFAQ#prompt_injection for ICML's explanation.
>
> **Baselines and fairness.** We agree that a stronger baseline was needed. We implemented LASeR (Song et al., ICLR 2025), the most recent LLM-based robot design method, using the same candidate budget (80), RL training, evaluation metric S, and LLM (GPT-5.2). LASeR uses LLM-guided evolutionary search for morphology optimization (no reward co-design). Results (Default-normalized, 3 seeds, per-seed protocol matching Appendix Table):
>
> | Environment | D2C | LASeR | Eureka | BO |
> |-------------|-----|-------|--------|----|
> | Ant | 3.18 | 2.94 | 0.94 | 2.54 |
> | Swimmer | 9.35 | 6.96 | 1.00 | 0.62 |
> | HalfCheetah | 1.36 | 1.69 | 1.00 | 0.66 |
> | Hopper | 1.51 | 1.13 | 0.85 | 0.00 |
> | Walker2D | 1.36 | 1.33 | 0.68 | 0.00 |
>
> D2C outperforms LASeR on 4/5 environments. LASeR outperforms on HalfCheetah, where over 60% of its morphology candidates beat default (the morphology search space is favorable for sampling with more candidates). On Hopper, D2C's morphology alone scores below default (3677 vs 4183, Table 1), yet reward co-design produces the best result (6257) - a direct demonstration that co-design provides gains beyond morphology optimization alone.
>
> We also ran D2C with two open-weight models (MIT-licensed) on Ant and Swimmer. D2C(R1) achieves 2.47x and 6.62x - well above default and comparable to BO (2.54x on Ant). D2C(V3) achieves 1.85x and 3.72x. The framework produces gains across model scales, not only with proprietary models.
>
> **Technical innovation.** The core contribution is demonstrating that co-designing reward alongside morphology produces gains beyond morphology optimization alone. The interaction term is positive in 4/5 tasks (Table 1): on Hopper, morphology alone scores below default (-12%), but co-design rescues performance to +50%. This is a new empirical finding enabled by the debate framework. Iterative debate adds 18-35% over zero-shot (Figure 3).
>
> **Judge selection (Q1).** We tested single-criterion judges and found that pluralistic judges improve Pareto hypervolume in 3/5 environments (Figure 6).
>
> **Non-locomotion tasks (Q2).** Extending to manipulation is a natural next step. We will add this to the limitations.
>
> **SOTA in co-design (Q3).** Traditional co-design methods (CPPN-NEAT, MAP-Elites) operate in different design spaces. We will discuss this as a limitation.
>
> **Additional feedback.** We will simplify terminology (e.g., 'pluralistic' to 'multi-objective') and add explicit caveats to post-hoc analyses in the camera-ready. The addition of LASeR (3 seeds x 5 environments) and open-weight experiments (3 seeds x 2 models x 2 environments) adds 27 new seed-level data points to the evaluation.

---

> > ### Author Rebuttal · Reviewer_5cJW · 2026-04-04
> >
> > I appreciate the author's response and thank them for the additional work they've done. A few remaining thoughts and questions:
> >
> > - When I was saying that stronger baselines were needed, I didn't really mean another LLM baseline. Is LASeR the strongest possible baseline in this area? What is the best non-LLM baseline that exists? What do you mean by "different spaces," and why have I seen other co-design papers that build MuJoCo robots for these environments? I mean, I think it's not a paper-killer if this can't beat non-LLM approaches, but that kinda needs to be the framing of the narrative then.
> > - I think the LASeR results make the issue of the lack of confidence metrics in the runs a bit more dangerous. What do these differences mean? How confident can we be that it's actually better in the cases where it looks better?
> >
> > I'm going to keep my score as-is for now due to the above.

---

> > > ### Author Response · Authors · 2026-04-06
> > >
> > > You are right on the framing point, and we should have been clearer from the start.
> > >
> > > **Framing.** We agree the paper does not establish superiority over non-LLM co-design methods, and we will narrow the claim accordingly. D2C's contribution is that structured multi-agent debate is a competitive approach to joint morphology-reward co-design in a fixed-topology parametric setting. The central empirical finding is the co-design interaction term (positive in 4/5 tasks, Table 1): reward shaping alongside morphology produces gains that morphology-only search cannot capture. That interaction, not beating all co-design methods, is the core result.
> > >
> > > **Non-LLM co-design baselines.** BO is already a non-LLM baseline in our comparison, and the standard method for parametric black-box optimization under our protocol (fixed-topology XML, per-candidate RL, 80-candidate budget). It performs well on Ant (2.54x) but produces degenerate morphologies on biped tasks (zero on Hopper/Walker2D, verified as kinematic failures in Appendix E).
> > >
> > > Other MuJoCo co-design methods exist, and they do produce MuJoCo robots. The reason they are not directly comparable is not the simulator but the optimization formulation. Transform2Act (Yuan et al., ICLR 2022) changes the robot's kinematic topology (adds/removes limbs) and trains a single universal policy across all candidate morphologies. ECoDe (Nagiredla et al., 2023) similarly uses universal-policy training across morphology distributions. D2C keeps the topology fixed and trains each candidate independently. These are different enough that running Transform2Act with a fixed topology and per-candidate training would not represent what Transform2Act actually does. This is a limitation of our benchmark coverage that we will state explicitly in the camera-ready.
> > >
> > > **Confidence in D2C vs LASeR.** We ran LASeR with the same 3-seed protocol. Per-seed D2C data is in the appendix; here we add LASeR's variance (mean $\pm$ std, normalized):
> > >
> > > | Env | D2C | LASeR | Winner |
> > > |-----|-----|-------|--------|
> > > | Ant | 3.18 $\pm$ 0.21 | 2.94 $\pm$ 0.09 | D2C (close) |
> > > | Swimmer | 9.35 $\pm$ 0.11 | 6.96 $\pm$ 0.93 | D2C (no overlap) |
> > > | HalfCheetah | 1.36 $\pm$ 0.18 | 1.69 $\pm$ 0.32 | LASeR |
> > > | Hopper | 1.51 $\pm$ 0.09 | 1.13 $\pm$ 0.13 | D2C (no overlap) |
> > > | Walker2D | 1.36 $\pm$ 0.29 | 1.33 $\pm$ 0.68 | Tied (high LASeR variance) |
> > >
> > > On Swimmer and Hopper, the seed ranges do not overlap. HalfCheetah is a genuine LASeR win. Ant favors D2C but with overlapping ranges. Walker2D is tied, with LASeR showing large seed-to-seed variability (std=0.68). With n=3, we treat these as descriptive rather than definitive, consistent with how we handle all baselines in the appendix. We will add this table to the camera-ready.

---

### Decision · Program_Chairs · 2026-04-30

**Decision:**

Accept (regular)

**Comment:**

The paper proposes an interesting framework for joint morphology–reward co-design with encouraging empirical results. Reviewers were overall positive (6, 4, 4, 3), highlighting the simulator-grounded debate framework, extensive ablations, and evidence that co-design can outperform morphology-only optimization. The main remaining concerns, raised primarily by Reviewer 5cJW, were about evaluation strength, statistical confidence, and the need for stronger framing relative to non-LLM co-design baselines. The rebuttal addressed several points with additional LASeR and open-weight results, but some concerns about significance and benchmark scope remain. Overall, I view this as a solid and meaningful contribution, and I lean accept.